# OptiFluence: Principled Design of Privacy Canaries

**Mohammad Yaghini** [1]  **Michael Aerni** [* 2]  **Junrui Zhang** [* 1]  **Nicolas Papernot** [1]  **Florian Tramèr** [2]

## Abstract

Privacy auditing has emerged as a practical tool for empirically estimating training data leakage in machine learning models, in contrast to the provable but often overly pessimistic bounds provided by differential privacy analysis. A common strategy is to use membership inference attacks to detect the presence of specific *canaries*—data points chosen to maximize attack success—in training data. However, existing canary designs are largely heuristic, relying on mislabeled or out-of-distribution samples. We address this gap by formulating canary design as a bilevel optimization problem, where the model is trained in the inner loop and the canary is optimized in the outer loop to maximize its detectability. To solve this problem, we develop OptiFluence, a scalable optimization framework that combines (i) initialization by selecting candidates using influence functions and (ii) unrolled optimization with memory-efficient techniques. Our approach achieves remarkable empirical performance on four datasets. Optimized canaries achieve near-perfect detection rates of 99.6% true positive rate at 0.1% false positive rate on `CIFAR-10`, outperforming in-distribution baselines by $4\times$. Critically, these canaries transfer effectively across different model architectures without retraining, enabling practical third-party privacy audits. This transferability allows regulators and auditors to assess model privacy without requiring access to proprietary training infrastructure or substantial computational resources.[1]

## 1. Introduction

Machine learning models can inadvertently leak sensitive information about their training data, raising significant privacy concerns in real-world applications (Carlini et al., 2021). Because it is difficult to obtain a realistic assessment of such privacy leakage in a practical ML pipeline using only natural data, researchers often insert canaries into training sets and then measure whether membership inference attacks (MIAs) can detect them.

Existing approaches to designing canaries, however, are largely heuristic and ad hoc. They typically rely on mislabeled examples (Steinke et al., 2023), out-of-distribution points (Meeus et al., 2025), or adversarial perturbations (Wen et al., 2022). While such strategies are simple to deploy, they risk underestimating privacy leakage: if the chosen canaries are not maximally detectable, the resulting audit may provide a false sense of security. Moreover, privacy auditing is not limited to models trained with differential privacy. Standard training pipelines can exhibit meaningful implicit privacy through mini-batch subsampling and the inherent stochasticity of SGD (Balle et al., 2018; Altschuler* & Talwar*, 2022). These sources often go unquantified in practice, making principled canary design equally important in non-differentially-private settings.

In this work, we present a *principled* approach to optimizing privacy canaries. We formulate canary construction as a *bilevel optimization problem* that explicitly maximizes the likelihood ratio for membership inference—moving beyond heuristic designs toward a rigorous mathematical foundation. This formulation allows us to construct canaries that are provably more detectable than existing approaches.

We instantiate this framework through OptiFluence, which combines two complementary strategies: *influence-based pre-selection* to identify promising initial canaries from natural data, and *gradient-based sample optimization* using unrolled training dynamics to fine-tune these candidates. To make this approach scalable, we develop memory-efficient techniques that take advantage of *rematerialization* (gradient checkpointing) and truncated backpropagation through time (Williams & Peng, 1990).

Our optimized canaries achieve remarkable performance improvements. On `CIFAR-10`, we achieve $4\times$ higher de-

---
[*]Equal contribution  [1]University of Toronto & Vector Institute, Toronto, Canada [2]ETH Zurich, Zurich, Switzerland. Correspondence to: Mohammad Yaghini <mohammad.yaghini@mail.utoronto.ca>.

*Proceedings of the 43rd International Conference on Machine Learning*, Seoul, South Korea. PMLR 306, 2026. Copyright 2026 by the author(s).

[1]Code available in https://github.com/cleverhans-lab/optifluence.

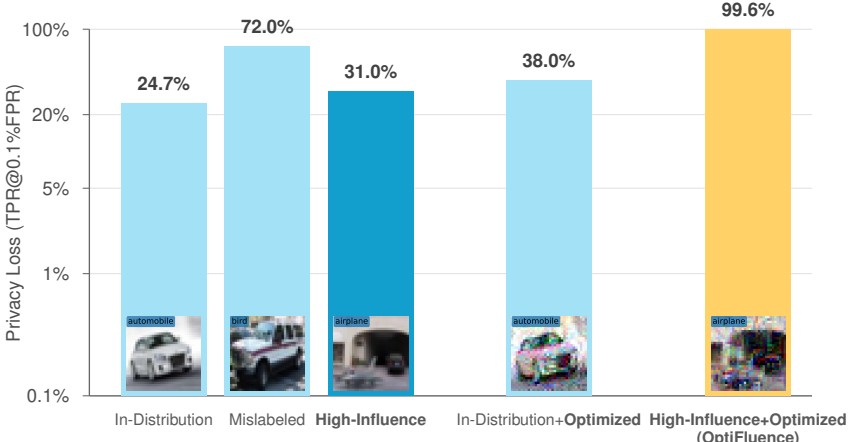

*Figure 1.* **OptiFluence yields canaries with almost perfect detectability.** Our canary pipeline involves two stages: initialization and optimization. OptiFluence (highlighted) picks the strongest primitives at each stage: a robust initialization based on influence (IF-Init) and a scalable optimization strategy (ReMat+TBPTT).

tectability than in-distribution baselines and near-perfect detection rates (99.6% TPR at 0.1% FPR; see Figure 1). Critically, these canaries transfer effectively across different model architectures—a property that enables practical *third-party auditing* without requiring access to the model provider's infrastructure or placing excessive computational demands on auditors such as regulators with limited resources.

This possibility of scalable third-party audits is a key advantage of input-space canaries. Auditors can use optimized canaries for *privacy reconnaissance*: screen many models efficiently by black-box querying (no infrastructure access required if canary inclusion is mandated), then escalate suspected violators to expensive tight audits with gradient access. Our canaries' transferability (Section 6.2) makes this practical—canaries optimized once will work across providers and architectures. This two-stage approach balances regulatory burden: most compliant models pass screening quickly, while verification resources focus on high-risk cases. In contrast, tight gradient-based audits (Nasr et al., 2023) require close access to the training procedure, making comprehensive screening infeasible.

**Our contributions are:**

- We cast canary design as a bilevel optimization problem with a formal privacy loss objective, providing a principled foundation that unifies and significantly outperforms prior heuristic constructions.

- We develop OptiFluence, a scalable optimization pipeline that solves the canary optimization problem through influence-based pre-selection and memory-efficient unrolled sample optimization with rematerialization and truncated backpropagation.

- We demonstrate empirically on four datasets (MNIST, CIFAR-10, CIFAR-100, and HAM10K, a skin lesions dataset) that optimized canaries quantify privacy leakage more accurately, achieving near-perfect detectability (99.6% TPR@0.1%FPR on CIFAR-10) compared to in-distribution baselines.

- We show that OptiFluence-optimized canaries transfer across model architectures and training procedures, enabling efficient regulatory (third-party) auditing.

## 2. Related Work

Prior work has shown that certain training samples are inherently more vulnerable to membership inference. For instance, Kulynych et al. (2022) found minority subpopulations to be disproportionately exposed, while Carlini et al. (2021) observed similar vulnerability among outliers. In vision models, Jagielski et al. (2020) created poisoned points to stress-test DP-SGD, and in language models, Carlini et al. (2019b) and follow-ups (Yue et al., 2023; Meeus et al., 2025; Panda et al., 2024) demonstrated that models can memorize synthetic "secrets" inserted into training text. More recent studies extend this line to large language models (LLMs), using identifiers, rare sequences, or synthetic text to probe memorization (Yue et al., 2023; Meeus et al., 2025; Panda et al., 2024). Together, these works highlight the existence of privacy-sensitive samples, but they rely on task-specific heuristics.

Beyond inputs, privacy canaries have also been constructed in parameter space. For example, Maddock et al. (2023) proposed weight-space and gradient-based canaries in federated learning, later generalized to randomized updates (Pillutla et al., 2023). Other studies in the federated setting examined unintended memorization of user-specific features (Thakkar

et al., 2020). These broaden the notion of canaries but still depend on domain knowledge and manual design.

Perhaps the closest works to ours are Nasr et al. (2023) and Boglioni et al. (2025). Nasr et al. (2023, Algorithm 3) construct input-space gradients by searching for a canary whose gradient is orthogonal to the average in-distribution model update. We discuss this principle and its connection to OptiFluence and our baselines in Appendix B. Concurrent work by Boglioni et al. (2025) uses metagradients (Engstrom et al., 2025) to calculate privacy canaries over training trajectories. Boglioni et al. (2025), however, are focused on single-run auditing (Steinke et al., 2023), which means many canaries get injected into the training run to simulate the effect of adding a single sample in more traditional MIA attacks. By contrast, we only optimize a single canary, do not assume any particular auditing setup *a priori*, and assume that the canary will be added to the training set and sampled randomly by the training algorithm. From a canary optimization perspective, OptiFluence subsumes the work of Boglioni et al. (2025) as a special case by choosing an in-distribution sample and using rematerialized gradients. Our ablations in Section 6.3 cover this special case as well. Yoon et al. (2024) separate the training-time canary from the inference-time query: they insert a fixed canary into training and separately optimize an adversarial sample used only to probe the trained models. By contrast, OptiFluence optimizes the canary that is inserted into training by differentiating through the full training trajectory, targeting privacy leakage at its source rather than improving the downstream query.

In summary, while prior work demonstrates the usefulness of canaries for auditing, their design remains largely heuristic and ad hoc. Our contribution is to cast canary construction as an optimization problem, unifying these disparate efforts into a principled framework that yields stronger membership inference attacks.

# 3. Background

## 3.1. Influence Functions

Influence functions, scaled by Koh & Liang (2017) to large ML models, identify the "most influential" training samples for a prediction. Let $D_{\text{train}} = \{z_i = (x_i, y_i)\}$ and $\mathcal{L}(\theta; z_i)$ be the per-sample loss. The trained model $\theta^*$ solves $\theta^* = \operatorname{argmin}_{\theta \in \mathbb{R}^d} \frac{1}{N} \sum_{i=1}^{N} \mathcal{L}(\theta; z_i)$. Assuming $\theta^*$ exists and is unique, consider upweighting sample $z_j$ by $\alpha$: $\theta^*(\alpha) = \operatorname{argmin} \frac{1}{N} \sum_i \mathcal{L}(\theta; z_i) + \alpha \mathcal{L}(\theta; z_j)$. By the Implicit Function Theorem, the influence of $z_j$ is

$$I_{\theta^*}(z_j) \triangleq \frac{d\theta^*}{d\alpha}\Big|_{\alpha=0} = -H^{-1} \nabla_\theta \mathcal{L}(\theta^*; z_j), \quad (1)$$

where $H = \nabla_\theta^2 \ell(\theta^*; D_{\text{train}})$ is the Hessian at $\theta^*$, and $\theta^*(\alpha) - \theta^* \approx \alpha I_{\theta^*}(z_j)$ for small $\alpha$. See Appendix A for details on cross-sample influence effects.

## 3.2. Differential Privacy

We characterize privacy leakage via the canonical notion of differential privacy (Dwork et al., 2006).

**Definition 1** (Differential Privacy). *A randomized mechanism $\mathcal{M}$ is $(\varepsilon, \delta)$-DP if for all datasets $D, D' \in \mathcal{D}$ differing in one datapoint and for all events $\mathcal{O}$:*

$$P[\mathcal{M}(D) \in \mathcal{O}] \leq e^\varepsilon P[\mathcal{M}(D') \in \mathcal{O}] + \delta.$$

In the above definition, $\delta \in (0, 1)$ can be thought of as a very small failure probability, and $\varepsilon > 0$ is an upper bound on the *privacy loss*. Smaller $\epsilon$ and $\delta$ correspond to stronger privacy guarantees. We focus on the information leakage of a randomized mechanism $\mathcal{M}$ that trains a model on a private dataset $D \in \mathcal{D}$. Importantly, the training algorithm $\mathcal{M}$ need not be explicitly differentially private like DP-SGD (Abadi et al., 2016). We can audit any randomized training algorithm, including the canonical SGD.

Since the two datasets $D, D'$ can only differ in a single sample (according to Definition 1), privacy auditors seek to find the sample that gives them the best odds of detecting a change in the output of the mechanism. To concretize this, let us forgo the probability of failure and set $\delta = 0$. The privacy parameter $\varepsilon$ bounds the *odds* of detecting a change:

$$\log \{P[\mathcal{M}(D \cup \{x\}) \in \mathcal{O}] \, / \, P[\mathcal{M}(D) \in \mathcal{O}]\} \leq \varepsilon, \quad (2)$$

where, without loss of generality, we took $x = D' \setminus D$ to be the sample difference between $D$ and $D'$. We evaluate the auditor's success using *membership inference attacks* (MIAs)—the canonical technique for empirical estimation of privacy leakage of ML models (Shokri et al., 2017; Jagielski et al., 2020). A common MIA involves the auditor training **shadow models** with full knowledge of what samples were in the training set (IN), and which were not (OUT). Armed with this knowledge, the auditor seeks to predict membership of a sample $x$ in a target model's training set and predict IN vs. OUT. Following best practices of reporting worst-case metrics (Carlini et al., 2022), we report the success of the auditor in terms of **TPR@0.1%FPR**. This is the true positive rate (predicting IN when the ground truth is IN) at the stringently low false positive rate (predicting IN when the ground truth is OUT) of 0.1%.

The success of the auditor establishes a lower bound on the left-hand side of Equation (2), i.e., a *lower bound on the privacy leakage for sample $x$*. In this work, we employ a strong MIA known as LiRA (Carlini et al., 2022) (see Algorithm 2 in Appendix D.5), which formalizes MIA as a hypothesis test of membership. In Section 4, we take advantage of this interpretation to produce a privacy loss function that we can optimize.

## 4. Canary Optimization as Likelihood Ratio Maximization

Let $\mathcal{T}(D)$ denote a training algorithm on dataset $D$, and let $(x, y) \sim \mathcal{D}$ be a sample drawn from the data distribution. Consider a dataset $D \sim \mathcal{D}^n$ sampled i.i.d. from $\mathcal{D}$. We define two distributions over trained model parameters:

$$Q_{\text{in}}(x, y) = \{\theta \leftarrow \mathcal{T}(D \cup \{(x, y)\}) \mid D \sim \mathcal{D}^n\}$$
$$Q_{\text{out}}(x, y) = \{\theta \leftarrow \mathcal{T}(D) \mid D \sim \mathcal{D}^n\}$$

The first distribution $Q_{\text{in}}(x, y)$ captures the randomness in model parameters when $(x, y)$ is included in training, while $Q_{\text{out}}(x, y)$ captures the distribution when $(x, y)$ is excluded.

**Membership inference as hypothesis testing.** Given a trained model with parameters $\theta$, an adversary performing membership inference must distinguish between two hypotheses:

$$H_0 : \theta \sim Q_{\text{out}}(x, y) \quad \text{vs.} \quad H_1 : \theta \sim Q_{\text{in}}(x, y).$$

By the Neyman-Pearson lemma, the most powerful test at significance level $\alpha$ rejects $H_0$ when the likelihood ratio

$$\Lambda(\theta; x, y) = \frac{p(\theta \mid Q_{\text{in}}(x, y))}{p(\theta \mid Q_{\text{out}}(x, y))} \tag{3}$$

exceeds a threshold, where $p(\theta \mid Q_b(x, y))$ is the probability density of $\theta$ under distribution $Q_b(x, y)$ for $b \in \{\text{in}, \text{out}\}$.

**From model parameters to predictions.** The likelihood ratio in Equation (3) is intractable, as it requires computing densities over high-dimensional parameter spaces. Instead, we work with a computable proxy: the model's predictions on $(x, y)$. We replace the intractable likelihood ratio of parameters with the log-likelihood ratio of the model's predicted probabilities:

$$\hat{\ell}_{\text{priv}}(x, y) = \log p(y \mid \theta_{D \cup \{(x,y)\}}, x) - \log p(y \mid \theta_D, x) \tag{4}$$

where $\theta_D$ and $\theta_{D \cup \{(x,y)\}}$ denote models trained on $D$ and $D \cup \{(x, y)\}$, respectively. Intuitively, if $(x, y)$ was in the training set, the model should assign higher probability to the correct label $y$.

**Logit rescaling for numerical stability.** $\hat{\ell}_{\text{priv}}$ is tractable but suffers from computational issues. For stability and to ensure approximately Gaussian distributions (crucial for parametric modeling in our optimization), we follow Carlini et al. (2022) and replace the log-likelihood with a more practical test statistic—the hinge loss.

Let $g(\theta; x) \in \mathbb{R}^{|\mathcal{Y}|}$ be the model's pre-softmax logits. The logit transformation $\phi(p) = \log(\frac{p}{1-p})$ can be rewritten as $\phi(p(y|\theta, x)) = g(\theta; x)_y - \log \sum_{y' \neq y} \exp(g(\theta; x)_{y'}).$[2]

---

[2]See Appendix C for the full derivation.

We further follow (Carlini et al., 2022) and replace the $\log \sum \exp$ term with a computationally simpler maximum. The resulting *measurement function* is the hinge loss: $f(\theta; x, y) \triangleq g(\theta; x)_y - \max_{y' \neq y} g(\theta; x)_{y'}$. This loss measures the margin by which the correct class logit exceeds the maximum incorrect class's. Replacing the log-likelihood with the hinge loss in Equation (4) yields our *computational privacy loss*:

$$\ell_{\text{priv}}(x, y) = f(\theta_{D \cup \{(x,y)\}}; x, y) - f(\theta_D; x, y).$$

Maximizing $\ell_{\text{priv}}(x, y)$ over $(x, y)$ produces the canary that is most distinguishable under a likelihood-ratio test.

Sampling a privacy canary, therefore, reduces to solving the bilevel problem

$$\max_{(x,y)} \ell_{\text{priv}}(x, y) = f(\theta_{D \cup \{(x,y)\}}; x, y) - f(\theta_D; x, y) \tag{5}$$

$$\text{s.t.} \quad \theta_{D \cup \{(x,y)\}} \in \arg\min_{\theta} \frac{1}{|D|+1} \sum_{z_i \in D \cup \{(x,y)\}} \mathcal{L}(\theta; z_i),$$

$$\theta_D \in \arg\min_{\theta} \frac{1}{|D|} \sum_{z_i \in D} \mathcal{L}(\theta; z_i).$$

**Remark 1.** To simplify optimization and avoid discrete gradients over the label space, in this work we consider canaries with correct labels and only optimize the canary $x$. We find this strategy to be sufficient in finding highly distinguishable canaries; our ablations (Section 6.3) show that combining mislabeling and canary optimization does not improve a canary's distinguishability.

With a concretized privacy loss definition in Equation (4), our iterative algorithm to optimize the privacy loss $\ell_{\text{priv}}$ follows in Algorithm 1. The simplicity of the algorithm belies its main difficulty, namely, calculating the canary gradient $\nabla_x \ell_{\text{priv}}(x, y)$.

The gradient step (aka, canary gradient) is $\nabla_x \ell_{\text{priv}}(x) = \nabla_x f(\theta_{D \cup \{x\}}; x) - \nabla_x f(\theta_D; x)$. The term $\nabla_x f(\theta_{D \cup \{x\}}; x)$ seeks to maximize the privacy measurement $f$. However, this depends on $x$ at both train- and inference-time. The term $\nabla_x f(\theta_D; x)$ tries to minimize the original model's privacy measurement on $x$. Since $\theta_D$ does not have $x$ as a training-time dependency, optimizing this quantity is very similar to finding an *adversarial example*. The core challenge in optimizing the privacy loss in Algorithm 1 is therefore twofold: choosing an appropriate initial canary $x$ and the calculation of the gradient $\nabla_x f(\theta_{D \cup \{x\}}; x)$. In the next section, we present a remarkably effective approach: choosing a *high-influence* (Section 3.1) training sample as initialization and optimizing it further using unrolled gradients.

**Algorithm 1** Iterative Canary Optimization

**Require:** Initial canary $(x, y)$, Training set $D_{\text{train}}$, Canary learning rate $\eta_c$
1: **for** $t \in [T]$ **do**
2:     Train model $\theta^*$ on $D_{\text{train}} \cup \{x\}$
3:     Estimate $\nabla_x \ell_{\text{priv}}(x, y)$
4:     Update the canary: $x = x + \eta_c \nabla_x \ell_{\text{priv}}(x, y)$
5: **end for**

## 5. OptiFluence: Initialize with Influence, Optimize with Unrolled

In this section, we introduce our canary optimization method OptiFluence, which comprises two steps: i) initialization of the canary using influence functions (Section 5.1), and ii) optimization of the canary by efficient differentiation through unrolled model updates (Section 5.2).

### 5.1. Initialization Through Zeroth-Order Optimization of Influence

Real-world data distributions often contain rare sub-modes or memorized outliers (Carlini et al., 2019a), which are difficult to generalize from; thus, as argued by Feldman (2021), they require memorization to achieve better model performance. We sample our initial canary from such rare points using influence functions (see Section 3), which quantify interactions among training samples. While heuristic, our approach is motivated by the conjecture proposed by Feldman & Zhang (2020) that samples with high self-influence are the rare points that lie in the long-tail distribution: because they cannot be explained by the rest of the training distribution, the network is forced to memorize them.

We define the self-influence metric that we use for this purpose as follows:

**Definition 2** (Normalized Self-Influence). *Assume we have trained a model on the training set $D_{train} = D \cup \{x_i\}$, we define sample $i$'s normalized self-influence as its self-influence divided by the largest amount of (cross-)influence every other sample has on $x_i$. Formally,*

$$r_i = \frac{I(x_i; x_i)}{\max_{x_j \in D} I(x_i; x_j)}. \tag{6}$$

*As in Section 3, $\alpha I(x_i; x_j)$ is the change in sample $x_i$'s loss if we up-weighted sample $x_j$ by $\alpha$ during training.*

We initialize our canary optimization with the maximizer of normalized self-influence: $x_c = \text{argmax}_{x_i \in D_{\text{train}}} r_i$. We dub this initialization strategy **Ifluence-based Initialization**, or **IF-Init** for short. Writing the maximum normalized self-influence as $\max_{x_i \in D_{\text{train}}} r_i = \max_{x_i \in D_{\text{train}} = D \cup \{x\}} \frac{I(x_i; x_i)}{\max_{x_j \in D} I(x_i; x_j)}$, we note the connection to the Neyman-Pearson statistic $\Lambda$ for the membership

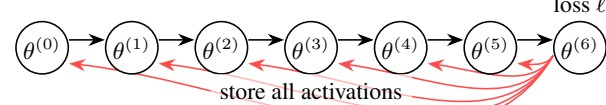

**(a) Unrolled: Full unroll & full backprop**

store all activations

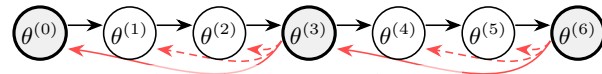

**(b) ReMat: Rematerialization (Checkpointing)**

store checkpoints; recompute between

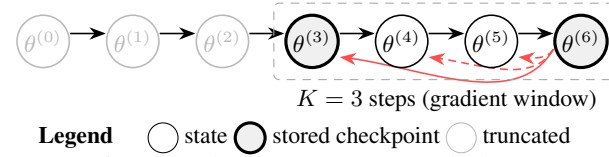

**(c) ReMat+TBPTT: Rematerialization truncated to a window of $K$ steps**

$K = 3$ steps (gradient window)

**Legend** ◯ state ◯ stored checkpoint ◯ truncated
→ forward → backprop ⇢ recompute+backprop

*Figure 2.* **Improving memory utilization in a trade-off with time (Rematerialization) and approximation (TBPTT).**

inference hypothesis test in Equation (3) and (after a $\log$) to the objective function in the canary optimization (5).

**Remark 2.** In maximizing the normalized self-influence, we restricted the measurement function to influence values over a discrete dataset $D_{\text{train}}$. Therefore, we are not optimizing over a compact set and thus cannot achieve the optima in problem (5). However, we use this pre-selection step to initialize our continuous canary optimization which does not have this limitation.

### 5.2. Implicit Differentiation Through Unrolled Model Updates

Consider the measurement function $f$, with the canary example $(x, y)$, where $\theta^{(0)} := \theta_D^*$ denotes the model obtained by training $\theta$ on dataset $D$ until convergence, and $\theta_{D \cup \{(x,y)\}}^*$ denotes the model obtained by continuing training on $D \cup \{(x, y)\}$. Then the privacy loss is defined as

$$\ell_{\text{priv}}(x, y) = f\left(\theta_{D \cup \{(x,y)\}}^*; x, y\right) - f(\theta_D^*; x, y).$$

For an optimization step, let $\psi := \theta_{D \cup \{(x,y)\}}^*$ and $\phi := \theta_D^*$. Since $\phi$ does not depend on $(x, y)$, expanding $\frac{df(\phi; x, y)}{dx}$ yields an explicit term $\frac{\partial f}{\partial x}\big|_\phi$ (with $\phi$ fixed) and an implicit term through $\phi$ that vanishes. The corresponding explicit term $\frac{\partial f}{\partial x}\big|_\psi$ appears in $\frac{df(\psi; x, y)}{dx}$ and cancels in the difference $\ell_{\text{priv}} = f(\psi; x, y) - f(\phi; x, y)$. The gradient therefore reduces to

$$\nabla_x \ell_{\text{priv}}(x, y) = \frac{df(\theta_{D \cup \{(x,y)\}}^*; x, y)}{dx} = \frac{\partial f}{\partial g} \cdot \frac{\partial g}{\partial \theta^*} \cdot \frac{d\theta^*}{dx},$$

where $\frac{\partial f}{\partial g}$ is the measurement gradient, and $\frac{\partial g}{\partial \theta^*}$ is the gradient of the canary-class logit, computable via a backward pass. The key issue is the term $\frac{d\theta^*}{dx}$, since $\theta^*_{D \cup \{x,y\}}$ is implicitly defined. In practice, we compute $\nabla_x \ell_{\text{priv}}$ directly via automatic differentiation, which correctly accounts for all terms. Here, instead, we *unroll the gradient updates* of the model after $T$ optimization steps as

$$\theta^*_{D \cup \{x,y\}} := \theta^{(0)} - \eta \sum_{t=0}^{T-1} \nabla_\theta \ell\big(\theta^{(t)}; D \cup \{(x,y)\}\big).$$

Replacing the training loss $\ell(\theta; D \cup \{(x,y)\})$ with $\frac{1}{n} \sum_i \mathcal{L}(\theta; x_i, y_i) + \mathcal{L}(\theta; x, y)$, derivative of the converged model w.r.t the input sample $\frac{d\theta^*}{dx}$ is

$$-\eta \sum_{k=0}^{T-1} \left[ \prod_{j=k+1}^{T-1} (I - \eta H^{(j)}) \right] \nabla_x \nabla_\theta \mathcal{L}\big(\theta^{(k)}; x, y\big), \tag{7}$$

where $H^{(j)}$ is the Hessian of the training loss at step $j$. Equation (7) makes the gradient tractable via automatic differentiation, but storing the full computational graph across all $T$ steps leads to prohibitive memory usage; Section 5.3 addresses this bottleneck (see Figure 2 for an illustration).

## 5.3. Re-Materialization with Truncated Backpropagation

The Jacobian $\nabla_x \nabla_\theta \mathcal{L}\big(\theta^{(k)}; x\big)$ in Equation (7) is the only term that directly depends on $x$; every other term depends on $x$ through previous parameter updates, so the entire computational graph must be retained. An alternative is to trade memory for extra computation by checkpointing and recomputing parts of the computational graph, a strategy known as **Rematerialization** (or ReMat for short). Instead of storing all intermediate states $\{\theta^{(t)}\}_{t=0}^T$, only a subset is kept; missing states are recomputed during backpropagation at the cost of additional forward passes.

**Truncated Backpropagation Through Time (TBPTT)** further addresses scaling by truncating gradient propagation (Williams & Peng, 1990). Rather than computing $\nabla_x \ell = \sum_{t=0}^{T-1} \frac{\partial \ell}{\partial \theta^{(T)}} \prod_{j=t+1}^{T} \frac{\partial \theta^{(j)}}{\partial \theta^{(j-1)}} \frac{\partial \theta^{(t)}}{\partial x}$, TBPTT approximates it with a shorter horizon $\nabla_x \ell \approx \sum_{t=T-K}^{T-1} \frac{\partial \ell}{\partial \theta^{(T)}} \prod_{j=t+1}^{T} \frac{\partial \theta^{(j)}}{\partial \theta^{(j-1)}} \frac{\partial \theta^{(t)}}{\partial x}$ where $K \ll T$. This reduces both compute and memory requirements while still providing informative gradient signals.

Together, rematerialization and TBPTT enable unrolled optimization over long training runs for non-trivial models.

## 6. Evaluation

We introduce our setup in Section 6.1, evaluate the effectiveness of OptiFluence in Section 6.2, and validate our design in Section 6.3 with a thorough ablation study.

### 6.1. Experiment Setup

**OptiFluence instantiation and baselines.** For all OptiFluence canaries, we initialize the canary by searching for the highest influence training samples as discussed in Section 5.1; we call this step IF-Init.

For the optimization step, we consider two methods for the evaluation of the canary gradient $\nabla_x \ell_{\text{priv}}$. Our main method, OptiFluence uses unrolled model updates with rematerialization and truncation, ReMat+TBPTT. Unrolled also employs unrolled updates but features no rematerialization or truncation. Due to its large memory footprint, our ablations featuring Unrolled is limited to MNIST.

**Evaluation setup.** We evaluate OptiFluence on four standard datasets for privacy auditing: CIFAR-10, CIFAR-100, HAM10K, and MNIST. HAM10K (Tschandl et al., 2018) contains 10,015 dermatoscopic images of skin lesions across 7 categories. As medical patient data, it represents a realistic privacy-sensitive scenario for auditing. We note that larger datasets are often practically infeasible, as privacy auditing can require training hundreds (or thousands) of models. On MNIST, we train small two-layer MLPs, and define OptiFluence as IF-Init followed by Unrolled. On CIFAR-10, CIFAR-100 and HAM10K, we use ResNet-9 (He et al., 2016). Transferability experiments use ResNet-9, 18, 50 (He et al., 2016) and WideResNet16-4 (Zagoruyko & Komodakis, 2016) models. Due to the larger model sizes, we use ReMat+TBPTT for the second OptiFluence step; we choose $K = 4$ as the window size for CIFAR-10 and HAM10K, and $K = 2$ for CIFAR-100.

**Auditing procedure.** We consider the traditional DP auditing of Jagielski et al. (2020): we train many victim models for every single canary while varying the canary's membership. We use the LiRA membership inference attack (Carlini et al., 2022) with the hinge loss, and we follow contemporary practices by reporting the attack's true positive rate (TPR) at a fixed low false positive rate (FPR) (see Figure 6 in Appendix E for ROC curves).

For training victim and shadow models, we use an oracle that randomly samples a training dataset from a fixed known base dataset $D$. Following Carlini et al. (2022), we ensure that each canary is a member of the training data in exactly half of the models. We train 20k victim models per canary for MNIST, and 128 victim models for CIFAR-10 and CIFAR-100. Unless explicitly mentioned, we run three random seeds for each experiment, and we use the same training procedure and architecture for OptiFluence, victim models, and shadow models.

**Baselines.** We use in-distribution (ID) examples (i.e., randomly sampled from the training dataset), mislabeled ex-

*Table 1.* OptiFluence yields almost perfect detectability on all datasets. TPR@0.1%FPR (%) for baselines and OptiFluence canaries.

| Methods | In-Distribution | Mislabeled | Adversarial | **OptiFluence** |
|---|---|---|---|---|
| MNIST | $0.1 \pm 0.0$ | $92.1 \pm 5.2$ | $97.8 \pm 1.6$ | $\mathbf{100.0 \pm 0.0}$ |
| CIFAR-10 | $24.6 \pm 19.8$ | $72.0 \pm 22.3$ | $89.9 \pm 7.6$ | $\mathbf{99.6 \pm 0.2}$ |
| CIFAR-100 | $68.2 \pm 25.3$ | $68.2 \pm 27.2$ | $72.9 \pm 22.1$ | $\mathbf{100.0 \pm 0.0}$ |
| HAM10K | $0.5 \pm 0.9$ | $1.0 \pm 1.8$ | $6.3 \pm 3.4$ | $\mathbf{100.0 \pm 0.0}$ |

*Table 2.* **Auditing DP-SGD on CIFAR-10 with OptiFluence canaries.** We report the global-threshold TPR (Aerni et al., 2024).

| Theoretical $\varepsilon$ | 0.5 | 1 | 2 | 6 | 8 |
|---|---|---|---|---|---|
| **TPR@0.1%FPR** (%) | 0.0 | 1.6 | 1.6 | 4.7 | 6.2 |

*Table 3.* **Privacy lower bounds $\hat{\varepsilon}_-$ for DP-SGD on MNIST.** Lower bounds are computed using Zanella-Beguelin et al. (2023) with $\delta = 10^{-5}$ as confidence intervals.

| Theoretical $\varepsilon$ | 0.5 | 1 | 2 | 6 | 8 |
|---|---|---|---|---|---|
| **TPR@0.1%FPR** | 0.10 | 0.13 | 0.26 | 0.31 | 0.32 |
| **Lower bound $\hat{\varepsilon}_-$** | (0, $\infty$) | (0, $\infty$) | (0, $\infty$) | (0.26, 0.93) | (0.35, 0.74) |

amples (Nasr et al., 2021; Steinke et al., 2023; Aerni et al., 2024), and adversarial examples (Nasr et al., 2021) as baseline canaries. We defer the exact training and optimization details to Appendix D.

### 6.2. Validation

**OptiFluence produces canaries with nearly optimal TPR@0.1%FPR.** As shown in Table 1, OptiFluence achieves almost perfect canary detectability on all datasets (see Table 8 in Appendix E.1 for corresponding global-threshold results), demonstrating significant performance enhancement compared to baselines. In particular, OptiFluence canaries are often orders of magnitude more detectable. This stands in contrast to mislabeled and adversarial examples; those can exhibit high TPRs, but only unreliably (as indicated by the large standard error on CIFAR-100 and CIFAR-10). Notably, models trained on CIFAR-100 yield higher privacy detectability across baselines due to being more prone to overfitting and the larger number of classes (as noted by (Shokri et al., 2017)). See Figure 1 and Appendix E for the full results. In contrast, for MNIST, models readily memorize atypical samples, so mislabeled and adversarial baselines achieve high detectability (92.1% and 97.8%, respectively) but with large variance.

**OptiFluence scales with larger datasets.** Table 6 in Section D.1 reports wall-clock time and peak VRAM usage for OptiFluence across datasets. OptiFluence scales from the small MNIST dataset to CIFAR-10 and CIFAR-100 while maintaining near-perfect performance. Without memory optimization, exact unrolled gradients (Section 5.2) cannot fit ResNet models in memory; ReMat+TBPTT enables scaling to ResNet-50, WideResNet, and datasets with larger output layers (CIFAR-100 has $10\times$ more classes than CIFAR-10). The optimization step peaks at 80GB VRAM on a single H100 GPU and can be further reduced by tuning the TBPTT window size $K$—our ablations (Figure 9 in Appendix E.7) show performance remains high even with aggressive truncation. The canary initialization step has a higher computational cost but is performed only once.

**Optimized canaries transfer between architectures.** Transferability of input-space canaries is important, because it enables efficient *third-party audits*: auditors can require model trainers to include these samples in their training data (possibly with a zero-knowledge proof of sampling (Shamsabadi et al., 2024)).

We hence evaluate the transferability of OptiFluence-optimized canaries by optimizing canaries for a ResNet-9 model (trained on CIFAR-10) and inserting them into the training data of ResNet-18, ResNet-50 and WideResNet16-4 models. Our best canaries optimized on the small model transfer nearly perfectly (100% TPR@0.1%FPR) to the larger models—despite differences in architecture and training procedures (e.g., learning rate scheduling for the larger models). In Appendix E.5, we further present transferability results for the Unrolled baseline over different-width MLPs on MNIST, where we observe similarly strong transferability despite architectural differences.

**DP-SGD auditing.** Our canaries are architecture- and training-agnostic; as such, we can use them to audit DP-SGD. We hence optimize canaries for ResNet-9 on CIFAR-10, and insert them into DP-SGD training with privacy parameter $\varepsilon \in \{0.5, 1, 2, 6, 8\}$, $\delta = 10^{-5}$, and clipping norm 1.0. We use a Rényi-DP accountant (dp-accounting) to calculate the noise scale $\sigma$.

Table 2 shows the results for our most sensitive canary. As expected, attack success improves with larger privacy budgets but is significantly diminished compared to non-privatized models. However, we note that no input-space canary (including ours) provides tight bounds for DP-SGD: the corresponding adversary model does not have access to the sampling procedure and cannot manipulate model updates, a requirement for tight auditing (Nasr et al., 2021). Annamalai & Cristofaro (2024) achieve tighter lower bounds via worst-case initialization, pre-training the model on auxiliary data to maximize gradient distinguishability of the target canary. This requires the model provider to use a specific weight initialization, which conflicts with our independent third-party auditing setup; requiring a provider to include a specific training sample is a narrower and more auditor-friendly obligation than dictating an initialization

| Method | Initialization | | | Optimization | Results |
|---|---|---|---|---|---|
| | ID | Mislabeled | IF-Init | Remat+ TBPTT | TPR@0.1% FPR (%) |
| In-Distribution | ✓ | ✗ | ✗ | ✗ | 24.6 ± 19.8 |
| Influence-Initialized | ✗ | ✗ | ✓ | ✗ | 31.01 ± 17.11 |
| OptiFluence w/o IF-Init | ✓ | ✗ | ✗ | ✓ | 38.02 ± 25.52 |
| OptiFluence w/ Mislabeled | ✓ | ✓ | ✓ | ✓ | 75.52 ± 19.99 |
| **OptiFluence** | ✗ | ✗ | ✓ | ✓ | **99.60 ± 0.19** |

*Table 4.* **Ablation of OptiFluence on CIFAR-10.** Influence-based initialization and optimization each improve canary detectability, and the combination achieves near-perfect detectability.

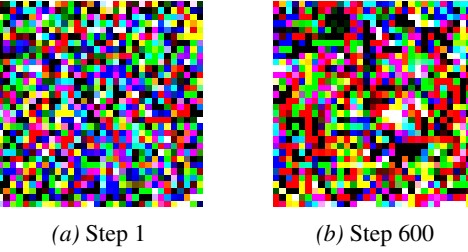

*(a)* Step 1       *(b)* Step 600

*Figure 3.* **Proper initialization is essential for truncated backpropagation.** A ReMat+TBPTT-optimized canary for CIFAR-10 initialized randomly achieves only 18.8% TPR@0.1%FPR. The canary looks very out-of-distribution.

strategy. Despite this, OptiFluence canaries remain detectable even under strong DP protection: at $\varepsilon = 1$, we achieve 1.6% TPR@0.1%FPR (see Table 2). Note that the non-DP baselines on CIFAR-10 (Table 1) are elevated due to memorization effects specific to this architecture and training setup; we discuss this in Section E.8.

**Privacy lower bounds.** While we report TPR@FPR directly, membership inference results can be converted to differential privacy lower bounds (Zanella-Béguelin et al., 2023): $\hat{\varepsilon}_- = \max\{\log \frac{1-\delta-\text{FPR}}{\text{FNR}}, \log \frac{1-\delta-\text{FNR}}{\text{FPR}}\}$, where FNR $= 1-$ TPR. We apply this to DP-SGD on MNIST (Table 3). However, for strong privacy guarantees ($\varepsilon \leq 2$), estimated bounds are vacuous despite non-zero TPR, and remain wide even for $\varepsilon \geq 6$. This occurs because the conversion requires large TPR/FPR differences and substantially more shadow models than our 20k for tight bounds in the low-$\varepsilon$ regime.

We thus report TPR@FPR as our primary metric: (i) it applies to any training algorithm (DP or non-DP); (ii) TPR@FPR remains informative even when $\hat{\varepsilon}_-$ is vacuous; and (iii) recent work shows privacy risks can be formalized directly in an attack framework without DP parameters (Kulynych et al., 2024).

### 6.3. Ablations

We perform ablation studies on CIFAR-10 to explain our design choices at each stage of OptiFluence. As summarized in Table 4, each choice is necessary to generate the strongest possible canary.

**Proper initialization is essential for truncated backpropagation.** TBPTT skips earlier steps of training, which introduces approximation errors in the unrolled gradient. Because of this error, canary initialization is crucial for ReMat+TBPTT. For example, TBPTT starting from a random initialization yields a canary with 18.8% TPR@0.1%FPR. Figure 3 hints at the cause: Despite showing distinct patterns compared to the initialization, the final canary does not resemble a CIFAR-10 image. This is in contrast to Unrolled-optimized canaries, which resemble in-distribution samples even for random initializations (see Figure 5). Hence, initialization from a high-influence training image (IF-Init) is necessary for achieving high detectability, thereby justifying OptiFluence's two-step design.

**With canary optimization, mislabeling is not effective.** Mislabeled samples are known to be strong canaries; indeed, we find that mislabeling in-distribution images increases their detectability. However, IF-Init retains the selected image's true label. Thus, one might expect that mislabeling this initial canary further improves detectability after optimization. Yet, as seen in Table 4 (OptiFluence + Mis.), mislabeling the high-influence sample before optimization *decreases* the TPR@0.1%FPR almost threefold.

Mislabeling creates label-inconsistent outliers that models must memorize (Zhang et al., 2021; Arpit et al., 2017). While memorized examples can be stronger canaries than typical samples, IF-Init already targets naturally hard-to-fit tail examples (and outperforms mislabeled baselines). We conjecture that mislabeling tail samples compounds two sources of anomaly (atypicality + incorrect label), making them too anomalous for our parametric membership inference approach to effectively detect.

**Truncated backpropagation introduces negligible error.** We use ReMat as an approximation where Unrolled optimization is computationally infeasible. As a reminder, ReMat+TBPTT truncates gradients to only a window of $K$ training steps, thereby introducing a tradeoff between computational cost and approximation errors. We empirically evaluate this tradeoff in Figure 9. The ReMat+TBPTT-optimized canaries match Unrolled-optimized ones in terms of TPR@0.1%FPR, despite the small window sizes ($K \in (2, 4)$). Moreover, we observe that the resulting surrogate canary loss is *lower under truncation*, suggesting that truncation with well-chosen window sizes can act as a regularizer.

The combined steps result in our final method OptiFluence, which achieves 99.6% TPR@0.1%FPR. We next situate these empirical gains within a theoretical framework by con-

necting OptiFluence to the gradient orthogonality condition used in prior work.

### 6.4. Discussion: Connection to the Gradient Orthogonality Condition

OptiFluence addresses the canary gradient $\nabla_x f(\theta_{D \cup \{(x,y)\}}; x, y)$ by differentiating directly through the unrolled training trajectory. An alternative approach is IF-Opt (Appendix A), which instead uses first-order derivatives of influence functions to approximate this gradient without unrolling. Concretely, IF-Opt approximates the effect of perturbing the canary $x$ on the measurement $f$ via:

$$
\begin{aligned}
\nabla_x f\big(\theta_{D \cup \{x\}}; x\big) \approx \\
-\nabla_\theta f\big(\theta_{D \cup \{x\}}; x\big)^\top \mathbf{H}^{-1} \nabla_x \nabla_\theta \mathcal{L}\big(\theta_{D \cup \{x\}}; x\big), \quad (8)
\end{aligned}
$$

where $\mathbf{H} = \nabla_\theta^2 \mathcal{L}(\theta^*; D_{\text{train}})$ is the Hessian of the training loss. The inverse Hessian $\mathbf{H}^{-1}$ is the central quantity: it accounts for how curvature of the loss landscape shapes the canary's influence on the trained model.

Maddock et al. (2023) independently arrive at an inverse-Hessian quantity as the principled canary objective via a different route. Under the assumption that model parameter updates $u$ follow a Gaussian distribution $\mathcal{N}(\mu, \Sigma)$, the log-likelihood ratio between models trained with versus without the canary $x$ evaluates to

$$
\log\left(\frac{p_{\text{in}}(u)}{p_{\text{out}}(u)}\right) = \pm \tfrac{1}{2} \nabla \ell(x)^\top (k\Sigma)^{-1} \nabla \ell(x),
$$

where $\Sigma$ is the covariance of updates and $k$ is the number of updates. Maximizing this expression requires computing the inverse-Hessian-vector product (IVHP) $(k\Sigma)^{-1} \nabla \ell(x)$, connecting directly to the $\mathbf{H}^{-1}$ computation in Equation (8). Maddock et al. declare computing $(k\Sigma)^{-1}$ in high dimensions infeasible, and instead approximate $\Sigma \approx \frac{1}{n} \sum_i u_i u_i^\top$, replacing the IHVP with the tractable proxy $\frac{1}{n} \sum_i \langle u_i, \nabla \ell(x) \rangle^2$. Nasr et al. (2023, Algorithm 3) pursue a related but separately motivated objective: minimizing the cosine similarity between the canary gradient and the mean in-distribution gradient. We show in Appendix B that both approaches implement variants of the same underlying principle, which we call the *orthogonality condition*, and that both treat the model as fixed, ignoring how canary inclusion changes the trained parameters.

Both IF-Opt and OptiFluence go beyond this fixed-model proxy by directly modeling how canary inclusion changes the trained model and, in turn, the privacy loss. This design choice already yields substantial gains: Nasr et al. (2023, Algorithm 3) achieves only 0.2–7.4% TPR@0.1%FPR on MNIST, matching heuristic baselines (see Appendix B). The two methods differ, however, in approximation quality. IF-Opt uses the EK-FAC Hessian approximation together with

the implicit function theorem, which introduces convergence assumptions. While the resulting zero-order self-influence signal is effective for candidate pre-selection (see IF-Init in Section 5.1), the first-order gradient in Equation (8) produces noisy, high-variance estimates that limit optimization performance. OptiFluence avoids these approximations entirely by differentiating directly through the unrolled training trajectory, yielding lower-variance gradient estimates that drive the rest of the performance gap visible in Table 5.

## 7. Conclusions

We introduced OptiFluence, a principled framework for privacy auditing that formulates canary design as a bilevel optimization problem. By combining influence-based preselection with memory-efficient unrolled optimization, our method systematically constructs canaries that maximize their detectability under membership inference attacks. This approach addresses the limitations of prior heuristic strategies, such as mislabeled or out-of-distribution samples, that may underestimate privacy leakage and yield inconsistent auditing signals.

Our empirical results across four datasets (MNIST, CIFAR-10, CIFAR-100, and HAM10K) demonstrate that optimized canaries achieve near-perfect detectability—99.6% TPR@0.1%FPR on CIFAR-10, $4\times$ higher than in-distribution baselines—while also transferring effectively across model architectures. These findings show that privacy auditing can move beyond ad hoc constructions toward a rigorous and scalable methodology.

**Limitations.** Optimized canaries may not resemble realistic training samples. Imposing a realism constraint via a similarity metric to the data distribution is straightforward within our bilevel framework, but requires committing to an acceptable input distribution, which itself has privacy implications we leave to future work.

Looking forward, we believe that optimized canaries can serve as standardized auditing primitives for both practitioners and regulators. They not only provide tighter empirical lower bounds on privacy leakage, but also establish a foundation for auditing frameworks that are model- and domain-agnostic. Future work should extend these ideas to larger datasets, federated and distributed settings, and auditing of differentially private training at scale. Ultimately, our results suggest that principled canary optimization is a key step toward reliable and reproducible privacy audits in modern machine learning.

## Impact Statement

This paper presents methods to improve privacy auditing of machine learning models. The primary intended impact is beneficial: enabling regulators, auditors, and model providers to more effectively assess privacy risks in deployed ML systems. By making privacy auditing more accessible and efficient, our work can help protect individuals whose data is used in model training.

As with any security or privacy research, there is potential for misuse. The techniques we present could inform adversaries about which types of training data are most vulnerable to privacy attacks, or help malicious actors craft data points specifically designed to be memorized. We believe the benefits of enabling effective privacy auditing substantially outweigh these risks, as transparent auditing mechanisms are essential for building trustworthy ML systems. Furthermore, the existence of optimized canaries incentivizes the development and deployment of stronger privacy protections, such as differential privacy, ultimately improving privacy outcomes.

We encourage responsible disclosure practices when using these methods for auditing production systems, and recommend that auditors work with model providers to remediate identified privacy risks rather than publicly exposing vulnerabilities.

## Statement on LLM Use

We have used LLMs for the purposes of a) re-writing and paraphrasing text in the paper for clarity; and b) coding and implementation of some of the techniques.

## Acknowledgement

We acknowledge the following sponsors, who support our research with financial and in-kind contributions: CIFAR through the Canada CIFAR AI Chair, NSERC through the Discovery Grant and an Alliance Grant with ServiceNow and DRDC, the Schmidt Sciences foundation through the AI2050 Early Career Fellow program. Resources used in preparing this research were provided, in part, by the Province of Ontario, the Government of Canada through CIFAR, and companies sponsoring the Vector Institute. We thank the computing team at the University of Toronto's Computer Science Department for administering and procuring the compute infrastructure used for the experiments in this paper.

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

# A. IF-Opt: First-order Optimization with Influence Functions

In this section, we present a new method to calculate the canary gradient using the model's inverse-Hessian-vector products (IHVPs). This formulation of the canary gradient allows us to benefit from the advances in calculation of IHVPs of large neural networks (e.g. EK-FAC approximations) which are primarily studied in the training data attribution literature.

## A.1. Background

Influence functions are a classical technique from robust statistics that were made scalable by Koh & Liang (2017) to explain large ML model's predictions by means of identifying the "most influential" training samples for a given prediction.

Assume we have trained the optimal model parameters $\theta^*$ on a training set $D_{\text{train}} = \{z_i \mid z_i = (x_i, y_i)\}$, where $x_i, y_i$ are input feature and labels. Using a *per-sample loss function* $\mathcal{L}(\theta; z_i)$ we have the classical ERM optimization problem $\theta^* = \operatorname{argmin}_{\theta \in \mathbb{R}^d} \frac{1}{N} \sum_{i=1}^N \mathcal{L}(\theta; z_i)$.

The classical influence setting assumes the optimizer $\theta^*$ exists and are unique. We want to understand the effect of adding (or removing) a new training example $z_c$ to the training set. We can parametrize the optimizer using the weight $\alpha$ of this sample: $\theta^*(\alpha) = \operatorname{argmin} \frac{1}{N} \sum_i \mathcal{L}(\theta; z_i) + \alpha \mathcal{L}(\theta; z_c)$ We define the *influence* of $z_c$ on $\theta^*$ via its first-order Taylor approximation of the response function at $\alpha = 0$. Given sufficient regularity, we can use the Implicit Function Theorem (Krantz & Parks, 2013) to calculate the influence as:

$$I_{\theta^*}(z_c) \triangleq \left. \frac{d\theta^*}{d\alpha} \right|_{\alpha=0} = -H^{-1} \nabla_\theta \mathcal{L}(\theta^*; z_c), \tag{9}$$

where $H = \nabla_\theta^2 \ell(\theta^*; D_{\text{train}})$ is the Hessian of the optimized model parameters on the training set. Using a first-order approximation of Equation (9) and setting $\alpha = \frac{1}{N}$, we can estimate the influence of changing sample $z_c$'s weight on the final model parameters with $\theta^*(\alpha) - \theta^* \approx \alpha I_{\theta^*}(z_c) = -\alpha H^{-1} \nabla_\theta \mathcal{L}(\theta^*; z_c)$.

Note that $\theta^*(\alpha) - \theta^* \in \mathbb{R}^D$, i.e., the influence lives in the (possibly very large) weight-space. To make influence more interpretable, it is customary to define a *measurement* function $f$ and instead ask: how does changing sample $z_c$'s weight changes the measurement, i.e., $f(\theta^*(\alpha)) - f(\theta^*)$?

In the data attribution literature, a good measurement function is often the validation loss or the logits for the query example $z_c$. Using the chain rule, calculating the influence on measurement function $f$ is straightforward:

$$f(\theta^*(\alpha)) - f(\theta^*) \approx \alpha I_f(z_c) = -\alpha \nabla_\theta f(\theta^*)^\top H^{-1} \nabla_\theta \mathcal{L}(\theta^*; z_c) \tag{10}$$

where $I_f(\cdot)$ is the influence on measurement $f$.

## A.2. Canary Optimization Using Influence Functions

Returning to our original goal of calculating the canary gradient $\nabla_x \ell_{\text{priv}}(x, y)$, our reformulation of the privacy loss (Equation (4) in Section 4) appears as self-influence in Equation (10) which gives us a functional tool to maximize the privacy loss $\ell_{\text{priv}}(x, y)$.

Concretely, we use Koh & Liang (2017, Section 2.2). It is shown that the impact of perturbing a training sample $x$ to $x + \Delta x$ on a test sample $x_c$ measurement (e.g. loss) $f(h(\theta_{D \cup \{x\}}; x_c))$ is approximated by $dI_{f, \theta_{D \cup \{x\}}}^\top \Delta x$ where

$$dI_{f, \theta_{D \cup \{x\}}}(x, x_c) = -\nabla_\theta f(h(\theta_{D \cup \{x\}}; x_c))^\top H_\theta^{-1} \nabla_x \nabla_\theta \mathcal{L}(h(\theta_{D \cup \{x\}}; x)) \tag{11}$$

This is maximized when we set $\Delta x = \kappa dI_{f, \theta_{D \cup \{x\}}}(x, x_c)$ for some positive scalar $\kappa$. We choose the test point to be $x_c = x$ to match the quantity we seek to optimize. Therefore, we have

$$\nabla_x f(h(\theta_{D \cup \{x\}}; x)) = dI_{f, \theta_{D \cup \{x\}}}(x, x)$$

Note that $\nabla_\theta \mathcal{L}(h(\theta_{D \cup \{x\}}; x))$ is a Jacobian (w.r.t. input $x$) calculated over the canary's per-sample gradient (w.r.t model parameters $\theta$). Therefore, the gradient is of size $O(n^2 d)$ where $n^2$ is the number of inputs, and $d$ is the number of model parameters. To make this calculation viable for larger models, we: i) calculate the inverse-Hessian-vector product (IVHF) $H^{-1} \nabla_\theta \mathcal{L}(\theta^*; x)$ using the EK-FAC approximations; and b) implement the Jacobian as a pure function allowing the use of automatic gradient (autograd). We offer more implementation details in Appendix A.3.

## A.3. Influence Function Details

We rely on a `functorch`-based version of `kronfluence` to calculate our influence functions. The importance of `functorch` is to generate influence as a pure function; enabling the calculation of the Jacobian in Equation (11). In this section, we briefly review concepts used in this library to measure influence functions over non-converged and non-convex models. Our treatment in this section is largely based on Grosse et al. (2023).

**Proximal Bregman Response Function.** In Section A.1 we discussed the limitations of influence functions. Bae et al. (2022) show that calculating influence functions on non-converged or non-convex models amounts to training the model not with the ERM loss but with a modified objective known as the proximal Bregman objective (PBO). The resulting response function is known as the proximal Bregman response function (PBRF):

$$\theta^s(\alpha) = \underset{\theta \in \mathbb{R}^D}{\arg\min} \frac{1}{N} \sum_{i=1}^{N} D_{\mathcal{L}_i}\left(h\left(\theta; x_i\right), h\left(\theta^s; x_i\right)\right) + \frac{\lambda}{2} \left\|\theta - \theta^s\right\|^2 + \alpha \mathcal{L}\left(\theta; z_c\right), \tag{12}$$

where $\theta^s$ are the final (but not necessarily converged) model weights $\hat{y}_i = h(\theta; x_i)$ is the prediction of the model on sample $x_i$. The $D_{\mathcal{L}_i}(\cdot, \cdot)$ denotes the Bregman divergence for the output space loss function: $D_{\mathcal{L}_i}(\hat{y}, \hat{y}^s) = \mathcal{L}_y(\hat{y}, y_i) - \mathcal{L}_y(\hat{y}^s, y_i) - \nabla_{\hat{y}} \mathcal{L}_y(\hat{y}^s, y_i)^\top (\hat{y} - \hat{y}^s)$.

Applying the implicit function theorem as before on the new response function, we can calculate the influence on the PBRF:

$$I_{\theta^s}\left(z_m\right) = \left.\frac{\mathrm{d}\theta^s}{\mathrm{d}\alpha}\right|_{\alpha=0} = -(G + \lambda\mathbf{I})^{-1} \nabla_\theta \mathcal{L}\left(z_c, \theta^s\right). \tag{13}$$

Here, $G$ denotes the Gauss-Newton Hessian (GNH), defined as $G = \mathbb{E}[J^\top H\hat{y}J]$, where $J = \mathrm{d}\hat{y}/\mathrm{d}\theta$ represents the Jacobian of the model outputs with respect to the parameters, and $H\hat{y}$ is the Hessian of the loss function with respect to the outputs. The expectation is taken over the empirical data distribution.

Crucially, the proximal Bregman objective (PBO) remains well-defined even in the case of overparameterized or partially trained neural networks. Unlike the full Hessian $\mathbf{H}$, the matrix $\mathbf{G}$ is guaranteed to be positive semidefinite, and the regularized version $\mathbf{G} + \lambda\mathbf{I}$ is strictly positive definite for any $\lambda > 0$. Prior studies (Bae et al., 2022; Grosse et al., 2023) have therefore adopted the damped Gauss-Newton Hessian $\mathbf{G} + \lambda\mathbf{I}$ as a surrogate in influence function computations—an approach we also follow in this work.

**EK-FAC for fast inverse-Hessian-vector products (IHVPs).** EK-FAC treats the curvature of each linear or convolutional layer as a Kronecker product of two much smaller empirical covariance matrices. If the layer's Jacobian $J \in \mathbb{R}^{B \times P}$ (batch $B$, parameters $P$) is written as $J = (A \otimes S)$, then the Gauss-Newton/Hessian block is approximated by

$$H \approx G_{\text{EK-FAC}} = \underbrace{\left(\frac{1}{B} A^\top A\right)}_{\Phi \in \mathbb{R}^{n \times n}} \otimes \underbrace{\left(\frac{1}{B} S^\top S\right)}_{\Psi \in \mathbb{R}^{m \times m}},$$

so $H^{-1}$ factorizes as $(\Phi^{-1} \otimes \Psi^{-1})$. In a practical implementation, we:

1. **Collect statistics.** After a forward–backward pass, accumulate
$$\Phi = \frac{1}{B} A^\top A, \qquad \Psi = \frac{1}{B} S^\top S;$$
where exponential moving averages keep the factors up-to-date at low cost.

2. **Add damping.** Ensure positive-definiteness and numerical stability:
$$\Phi \leftarrow \Phi + \lambda I, \quad \Psi \leftarrow \Psi + \lambda I.$$

3. **Invert factors.** (once per update) Small sizes ($n, m \leq$ few hundred) let us do an exact Cholesky/SVD solve $\Phi^{-1}$, $\Psi^{-1}$.

4. **Form IHVP for any vector (v).** Reshape the layer slice of $v$ into matrix form $V \in \mathbb{R}^{n \times m}$. Apply the Kronecker inverse by two cheap solves:
$$\text{vec}\left(\Phi^{-1} V \Psi^{-1}\right) = \left(\Phi^{-1} \otimes \Psi^{-1}\right) v \approx H^{-1} v.$$

To calculate the IHVP for the whole-network, we repeat steps 1–4 for every layer and concatenate the results; no conjugate-gradient iterations are needed and each matrix–vector product costs $O(n^3 + m^3 + nm)$, independent of parameter count.

| Method | Initialization | | Optimization | | | TPR@0.1%FPR (%) | |
| --- | --- | --- | --- | --- | --- | --- | --- |
| | ID | Mislabeled | IF-Init | IF-Opt | Unrolled | ReMat+TBPTT | MNIST | CIFAR-10 |
| In-Distribution | ✓ | ✗ | ✗ | ✗ | N/A | ✗ | $0.13 \pm 0.02$ | $24.6 \pm 19.8$ |
| IF-Init | ✗ | ✗ | ✓ | ✗ | N/A | ✗ | $32.94 \pm 16.18$ | $31.01 \pm 17.11$ |
| Influence-Optimized | ✗ | ✗ | ✓ | ✓ | N/A | ✗ | $0.17 \pm 0.02$ | $1.00 \pm 0.85$ |
| OptiFluence w/o IF-Init | ✓ | ✗ | ✗ | ✗ | N/A | ✓ | $33.53 \pm 26.77$ | $38.02 \pm 25.52$ |
| OptiFluence + Mislabeled | ✓ | ✓ | ✓ | ✗ | N/A | ✓ | $72.72 \pm 22.24$ | $75.52 \pm 19.99$ |
| OptiFluence | ✗ | ✗ | ✓ | ✗ | N/A | ✓ | $\mathbf{100.00 \pm 0.00}$ | $\mathbf{99.60 \pm 0.19}$ |
| Unrolled | ✗ | ✗ | ✓ | ✗ | ✓ | ✗ | $99.60 \pm 0.07$ | N/A |

*Table 5.* **Ablation study of OptiFluence components on CIFAR-10** reproduced from Table 4 for comparisons with IF-Opt.

### A.4. Empirical Validation: Unrolled optimization beats influence optimization

We use influence functions to select strong initial canaries in the IF-Init step; hence, optimizing a canary's *influence* seems like a natural way to obtain strong canaries. However, in Table 5—reproduced for convenience from Table 4 now including IF-Opt results—we find that this yields subpar canaries: While a canary optimized with IF-Opt (starting with IF-Init) significantly outperforms the baselines, simply using IF-Init for initialization without further steps yields a higher TPR@0.1%FPR.

We argue that the causes are mismatched assumptions of influence functions, which IF-Opt optimizes over. First, this optimization relies on the EK-FAC approximation of the inverse Hessian for the Inverse Hessian-vector products (Grosse et al., 2023). EK-FAC assumes layer independence; although the approximation is theoretically upper bounded for deeper networks with skip-connections (e.g., ResNets), the approximation error remains unknown. Furthermore, influence functions also assume convergence and convexity (Hammoudeh & Lowd, 2024). Combined, these assumptions introduce variance and noise, as evidenced in a standard error as high as 2.25% against a TPR of 4.17%.

In contrast, Unrolled and ReMat evaluate the training data influence by differentiating through the entire training trajectory, sidestepping the convergence assumption. Since those procedures do not rely on the Implicit-Function Theorem, there are no curvature approximations, such that Unrolled and ReMat yield an exact gradient of the measurement (hinge loss) with respect to the training data (canary). Therefore, optimizing over these exact gradients produces much stronger result with lower variance.

## B. The Orthogonality Condition: An Extensive Comparison with Maddock et al. (2023) and Nasr et al. (2023)

In Maddock et al., Algorithm 1, we note the optimization is over the canary sample $z$ which happens in the input-space through minimization of the following canary loss:

$$\min_{z \in \mathbb{R}^d} \mathcal{L}(z) = \sum_i \langle u_i, C \cdot \nabla_\theta \ell(z) \rangle^2 + \max(C - \|\nabla_\theta \ell(z)\|, 0)^2 \qquad \text{(Line 5)}$$

Note, however, calculating the above loss, requires a "pool of clients" that send model updates $u_i$ (Line 2); therefore crafting the canary $z$ requires a federated set-up with a pool of model updates are available (which is not a requirement in our setting).

**Weight-space vs. input-space.** We can write the above loss $\mathcal{L}(z_t)$ re-factorized in terms of a canary gradient $u_c \triangleq \nabla_\theta \ell(z_t)$:

$$\min_{u_c \in \Theta} \tilde{\mathcal{L}}(u_c) = \sum_i \langle u_i, C \cdot u_c \rangle^2 + \max(C - \|u_c\|, 0)^2.$$

While we certainly *can* optimize canaries in the input-space $\mathbb{R}^d$, we can do so directly in the space of model parameters $\Theta$. Consequently, the effective threat model in Maddock et al. is weight-space gradients. This is supported by the pipeline in their Figure 2 where we clearly see that the adversary releases the update canary $u_c$ and not the canary sample $z$.

Nasr et al. (2023, Algorithm 3) implements a canary loss that seeks to "align" the canary gradient with the average in-distribution gradient $\vec{g}_{\text{dist}}$:

$$\min_{(x,y)} l_{\text{adv}}(x,y) = \left| \frac{\nabla l(\theta,(x,y)) \cdot \vec{g}_{\text{dist}}}{||\nabla l(\theta,(x,y))|| \, ||\vec{g}_{\text{dist}}||} \right| \text{ where } \vec{g}_{\text{dist}} = \frac{1}{|D|} \sum_{(x_i,y_i) \in D} \nabla l(\theta,(x_i,y_i)) \tag{14}$$

Unlike CANIFE, the threat model here indeed is the release of input-space canaries. We share results using our implementation of Algorithm 3 on `MNIST`.

**Empirical Results using Nasr et al. (2023, Algorithm 3).** We optimize the objective (14) down to $\ell_{\text{adv}} \leq 0.0001$. We then evaluate the resulting canaries using the same empirical setup as in Section 6.1 but using 5,000 shadow models:

- Initializing the in-distribution sample (following Line 4 of the algorithm) we achieve, 0.2% TPR@0.1%FPR.

- Initializing from a canary sampled uniformly at random, and optimizing using Algorithm 3, we achieve 7.4% TPR@0.1%FPR.

These results are close to in-distribution and adversarial example baselines for `MNIST` in Table 1, respectively. What these baseline share is the fact that the canary gradients are not being shaped by the model training dynamics. *The average in-distribution gradient $\vec{g}_{\text{dist}}$ is essentially a constant.*

Comparing this to our bilevel objective formulation in Equation (5) (reproduced here for convenience):

$$\max_{(x,y)} \ell_{\text{priv}}(x,y) = f(\theta_{D \cup \{(x,y)\}}; x,y) - f(\theta_D; x,y) \tag{re.5}$$

$$\text{s.t.} \quad \theta_{D \cup \{(x,y)\}} \in \arg\min_{\theta} \frac{1}{|D|+1} \sum_{z_i \in D \cup \{(x,y)\}} \mathcal{L}(\theta; z_i), \quad \theta_D \quad \in \arg\min_{\theta} \frac{1}{|D|} \sum_{z_i \in D} \mathcal{L}(\theta; z_i),$$

We note that the first constraint depends on the canary that is being put in the training set, therefore, a change in the canary $(x,y)$ leads to changes in the resulting model $\theta_{D \cup \{(x,y)\}}$ that needs to be taken into consideration in the unrolled loss. Nasr's $\ell_{\text{adv}}$ does not take the impact of the inclusion of the canary in the training procedure directly as we do, but rather approximates it by optimizing the proxy objective $\ell_{\text{adv}}$.

Nasr et al. justify their choice of the proxy objective empirically, noting in Section B.2 "using the dot product between the privatized gradient and the canary gradient is a sufficient metric for auditing DP-SGD." For the sake of the presentation, let us call this the "orthogonality condition."

To the best of our knowledge Maddock et al. 2023, first derived and justified the orthogonality condition. Notably, in Appendix A, authors connect their objective to the likelihood ratio test and derive the condition under the assumption that "model update follow Gaussian distribution $\mathcal{N}(\mu, \Sigma)$. The sum of $k$ model updates then either follows $\mathcal{N}(k\mu, k\Sigma)$ (without the canary) or $\mathcal{N}(k\mu + \nabla\ell(z), k\Sigma)$ (with the canary), recalling that $u_c \propto \nabla\ell(z)$." Under this assumption, Maddock et al. derive the likelihood ratio test between the null $p_0$ and alternative $p_1$ distributions:

> "
>    In particular for the centers of the Gaussian with and without the canary, $u \in \{k\mu, k\mu + \nabla\ell(z)\}$ ,
>
> $$\log\left(\frac{p_1(u)}{p_0(u)}\right) = \pm\frac{1}{2}\nabla\ell(z)^T(k\Sigma)^{-1}\nabla\ell(z).$$
>
> Maximizing this term will thus help separate the two Gaussians. **However, doing this directly is infeasible as it requires to form and invert the full covariance matrix $\Sigma$ in very high dimensions.** Instead, we propose to minimize $z \mapsto \left(\nabla\ell(z)^T\right)\Sigma(\nabla\ell(z))$ as it is tractable and can be done with SGD. Note that for sample model updates $\{u_i\}$ we can estimate the (uncentered) covariance matrix as $\frac{1}{n}\sum_i u_i u_i^T$ and thus
>
> $$\left(\nabla\ell(z)^T\right)\Sigma(\nabla\ell(z)) \approx \frac{1}{n}\sum \nabla\ell(z)^T\left(u_i u_i^T\right)\nabla\ell(z) = \frac{1}{n}\sum \langle u_i, \nabla\ell(z)\rangle^2.$$
>
> "

We see that **the key approximation that leads to the orthogonality condition is the inverse-product-Hessian (IHVP)** $\nabla\ell(z)^T(k\Sigma)^{-1}\nabla\ell(z)$ **which Maddock et al. consider infeasible to do with SGD. But this IHVP calculation is exactly what we do efficiently** using influence function in Appendix A (for the IF-OPT baseline—see Eq. 10) and improve upon using unrolled gradients in the main matter for OptiFluence!

## C. Privacy Loss Details

### C.1. Derivation of the Rescaled Logits

Let $g(\theta; x) \in \mathbb{R}^{|\mathcal{Y}|}$ denote the pre–softmax logits of a model with parameters $\theta$ on input $x$. The corresponding softmax probabilities are

$$p(y \mid \theta, x) = \frac{\exp\big(g(\theta; x)_y\big)}{\sum_{y' \in \mathcal{Y}} \exp\big(g(\theta; x)_{y'}\big)}.$$

We define the logit rescaling function

$$\phi(p) = \log\left(\frac{p}{1-p}\right).$$

To express $\phi(p(y \mid \theta, x))$ in terms of logits, observe that

$$1 - p(y \mid \theta, x) = \frac{\sum_{y' \in \mathcal{Y}} \exp\big(g(\theta; x)_{y'}\big) - \exp\big(g(\theta; x)_y\big)}{\sum_{y' \in \mathcal{Y}} \exp\big(g(\theta; x)_{y'}\big)} = \frac{\sum_{y' \neq y} \exp\big(g(\theta; x)_{y'}\big)}{\sum_{y' \in \mathcal{Y}} \exp\big(g(\theta; x)_{y'}\big)}.$$

Therefore,

$$\frac{p(y \mid \theta, x)}{1 - p(y \mid \theta, x)} = \frac{\exp\big(g(\theta; x)_y\big)}{\sum_{y' \neq y} \exp\big(g(\theta; x)_{y'}\big)}.$$

Then the re-scaled logits can be expressed as:

$$\phi(p(y \mid \theta, x)) = \log\frac{p(y \mid \theta, x)}{1 - p(y \mid \theta, x)} = g(\theta; x)_y - \log\sum_{y' \neq y} \exp\big(g(\theta; x)_{y'}\big).$$

The second term is a LogSumExp, which is a smooth approximation of the maximum function:

$$\log\sum_{y' \neq y} \exp\big(g(\theta; x)_{y'}\big) \approx \max_{y' \neq y} g(\theta; x)_{y'}.$$

Consequently,

$$\phi(p(y \mid \theta, x)) \approx g(\theta; x)_y - \max_{y' \neq y} g(\theta; x)_{y'}.$$

*Table 6.* Wall-clock time and peak VRAM usage measured for the canary initialization and optimization steps of OptiFluence and baseline methods on `MNIST` with simple MLPs, `CIFAR-10` with ResNet-9, and `CIFAR-100` with ResNet-9.

|  | Initialization Step | | Optimization Step | |
| --- | --- | --- | --- | --- |
|  | Wall-clock Time (s) | Peak VRAM Usage (GB) | Wall-clock Time (s) | Peak VRAM Usage (GB) |
| `MNIST` | 2,641 | 2.3 | 32 | 79 |
| `CIFAR-10` | 7,456 | 38 | 1,128 | 80 |
| `CIFAR-100` | 7,458 | 38 | 1,292 | 80 |

Therefore, the re-scaled logits provide the hinge-style loss used as our *measurement function*.

# D. Experiment Details

## D.1. Computational Efficiency Comparisons: VRAM usage and Wall-clock Runtimes

We compare the wall–clock time and peak VRAM usage of OptiFluence across all datasets. As shown in Table 6, OptiFluence scales from the small `MNIST` dataset to `CIFAR-10` and `CIFAR-100`, while maintaining near perfect performance. The optimization step peaks at 80GB of VRAM usage on a single H100 GPU, which can be further reduced by decreasing the TBPTT window size $K$ without degrading performance. Adjusting other hyper-parameters, such as batch size or learning rate, can also reduce the runtime, which may introduce some trade–offs in accuracy. Note that `CIFAR-100` is more complex than `CIFAR-10` and therefore requires more training epochs, which impacts computational cost. Nevertheless, our method still achieves a significantly high TPR@0.1%FPR, and with a configuration that maintains comparable computational costs as `CIFAR-10`. The canary initialization step requires longer computation, but it is performed only once, and IF-Init ensembles 20 models for robustness, contributing to the increased runtime; the per–model runtime is approximately 373s. Safely reducing the ensemble size to 5 offers a substantial runtime reduction with minimal effect on final canary selection performance.

## D.2. Optimization and Training Hyperparameters

If not mentioned otherwise, all experiments use the exact same model architecture and training procedure when optimizing canaries and when training models for auditing. For all `MNIST` experiments, we use a small MLP with two layers and a hidden dimension of 20 features. We train those MLPs using SGD with a learning rate of 0.1 and momentum 0.9 with batch size 128 for 10 epochs. On `CIFAR-10`, we use hyperlightspeedbench (Balsam, 2023) CNNs with default training hyperparameters if not mentioned otherwise, for ReMat+TBPTT, we use ResNet9.

**Architecture hyperparameters for transferability experiments.** Table 7 summarizes the key differences in training procedure across the architectures used in the cross-architecture transfer experiments (Section 6.2). Canaries are optimized on ResNet-9 and transferred to the other architectures without any re-tuning.

*Table 7.* Training hyperparameters for each architecture in the transferability experiments. Canaries optimized on ResNet-9 are inserted into the other models without modification.

| Architecture | Optimizer | LR | LR Schedule | Label Smooth. | Epochs | Role |
| --- | --- | --- | --- | --- | --- | --- |
| ResNet-9 | SGD | 0.1 | Cosine | 0.0 | 50 | Canary source |
| ResNet-18 | SGD | 0.1 | Cosine | 0.1 | 50 | Transfer target |
| ResNet-50 | SGD | 0.1 | Cosine | 0.1 | 50 | Transfer target |
| WideResNet16-4 | SGD | 0.1 | Cosine | 0.1 | 50 | Transfer target |

## D.3. Canary Initialization

We used influence functions (Kronfluence) to compute influence scores in Equation (6). On `MNIST`, we used the same architecture as for canary optimization and model training. On `CIFAR-10`, we used the HyperLightSpeedBench CNN for efficiency. To improve robustness, we averaged our metric over an ensemble of 20 independently trained models before sorting and selecting the canaries. For IF-Init, we selected three canaries from the top five ranked by the metric for each of the three trials in every experiment.

### D.4. Canary Optimization Implementation Details

**Unrolled optimization.** We reduce variance in unrolled optimization by calculating gradients over multiple models simultaneously. Concretely, at each step, we train two "IN" models on a dataset that contains the canary and two "OUT" models on a dataset without the canary. Next, we calculate the mean hinge score of all "IN" models and all "OUT" models. We then maximize the difference using gradient ascent with a learning rate of 1.0 and momentum 0.9 for 300 steps. Every initial canary is a uniformly random image.

**Influence optimization.** In contrast to unrolled optimization, influence optimization uses an existing training sample (that will be replaced) as the initial canary. Optimization then repeatedly trains a model and maximizes the canary's self-influence on that model using gradient ascent. We estimate each canary's self-influence using the full base dataset (which is in the threat model of having access to the data distribution). For `MNIST` models, we use a canary learning rate of 0.001, no momentum, and 50 update steps; for `CIFAR-10`, we use a canary learning rate of 0.001, momentum 0.9, and 100 update steps.

In both cases, we use early stopping to combat high variance in the optimization procedure. At each optimization step, we calculate the canary's hinge score on a model that was trained on the canary to the model that was trained on the initial canary from the base dataset; we use the canary with the largest gap in scores.

### D.5. Audit Setup

---

**Algorithm 2** LiRA Membership Inference Audit (for one sample $x$)

---

**Require:** Base dataset sampler $\mathcal{O}$, candidate sample $x$, shadow counts $N_{\text{in}}$, $N_{\text{out}}$ (typically equal), victim model $h_v$, score function $s(h, x)$ (e.g., hinged scaled logit), evaluation grid of thresholds $\mathcal{T}$

1:
2: **Shadow training**:
3: **for** $i = 1, \ldots, N_{\text{in}}$ **do**
4:      Draw $D_i \sim \mathcal{O}$ and set $D_i^{\text{in}} = D_i \cup \{x\}$
5:      Train shadow model $h_i^{\text{in}}$ on $D_i^{\text{in}}$
6: **end for**
7: **for** $j = 1, \ldots, N_{\text{out}}$ **do**
8:      Draw $D_j' \sim \mathcal{O}$ and set $D_j^{\text{out}} = D_j'$
9:      Train shadow model $h_j^{\text{out}}$ on $D_j^{\text{out}}$
10: **end for**
11: **Score aggregation**:
12: $S_{\text{in}} \leftarrow \{ s(h_i^{\text{in}}, x) \mid i = 1..N_{\text{in}} \}$
13: $S_{\text{out}} \leftarrow \{ s(h_j^{\text{out}}, x) \mid j = 1..N_{\text{out}} \}$
14: **Fit score models (Gaussian LiRA)**:
15: Fit $\mathcal{N}_{\text{in}} = \mathcal{N}(\mu_{\text{in}}, \sigma_{\text{in}}^2)$ to $S_{\text{in}}$
16: Fit $\mathcal{N}_{\text{out}} = \mathcal{N}(\mu_{\text{out}}, \sigma_{\text{out}}^2)$ to $S_{\text{out}}$
17: **Victim evaluation**:
18: Obtain victim score $z_v \leftarrow s(h_v, x)$
19: Compute log-likelihood ratio

$$\text{LLR}(x) = \log p_{\mathcal{N}_{\text{in}}}(z_v) - \log p_{\mathcal{N}_{\text{out}}}(z_v)$$

20: **Audit metric**:
21: **for** $\tau \in \mathcal{T}$ **do**
22:      Predict "member" if $\text{LLR}(x) \geq \tau$, else "non-member"
23: **end for**
24: Compute ROC and report TPR at desired FPR (e.g., 0.1%)

---

For all results, we perform LiRA-style (Carlini et al., 2022) membership inference with 128 shadow models and the hinge score. That is, we first train shadow models while ensuring that each sample is a member in exactly half of them. Then, for every canary, we fit a Gaussian distribution on all member and non-member scores, and use their log-likelihood ratio on a victim model's score as the attack statistic. Similarly, we ensure that every sample is a member of exactly half of the victim

models.

We calculate ROC scores over predictions from victim models by using membership as the label, and we again ensure that each canary is a member in exactly half of the models' training data. Whenever we audit multiple canaries in parallel, we concatenate all their attack scores (following Aerni et al. (2024)).

As mentioned in Section 6.1, we consider two auditing setups: *traditional DP auditing* and an *efficient auditing heuristic*. Given the aforementioned membership inference procedure, those two setups only differ in the number of victim and shadow models. Concretely, we use 20k victim models for a single canary (resulting in 20k predictions) in the traditional setup, and the heuristic setup uses 512 victim models with 32 canaries (resulting in around 16k predictions). The heuristic setup allows us to obtain auditing results for a broader set of settings at the cost of slightly underestimated worst-case privacy leakage (Aerni et al., 2024).

### D.6. Baseline Canaries

We use three types of baseline canaries: in-distribution samples, mislabeled samples, and adversarial examples. For in-distribution baselines, we simply use the original image and label of the sample that would be replaced with a canary. For mislabeled samples, we use the same images, but flip their label to a different one selected uniformly at random. Lastly, we optimize adversarial examples using the FGSM attack (Goodfellow et al., 2015) with an epsilon of 0.3 and the $\ell_\infty$-norm. To improve robustness of the resulting adversarial examples, we maximize the minimum loss over 8 models.

## E. Additional Experiments

### E.1. Global-Threshold Results

Table 8 reproduces the main results of Table 1 using a global score threshold rather than per-sample LiRA thresholds (Aerni et al., 2024). OptiFluence achieves perfect detectability under both evaluation protocols, confirming that the strong performance in the main paper is not an artifact of the per-sample thresholding procedure.

*Table 8.* **OptiFluence yields almost perfect detectability under global thresholding.** Global-threshold TPR@0.1%FPR (%) for baselines and OptiFluence canaries (MLP for MNIST and ResNet-9 otherwise).

| Dataset | In-Dist | Mislabeled | Adversarial | OptiFluence |
|---|---|---|---|---|
| MNIST | $0.16 \pm 0.02$ | $92.13 \pm 5.17$ | $97.79 \pm 1.60$ | $\mathbf{100.00 \pm 0.00}$ |
| CIFAR-10 | $0.21 \pm 0.09$ | $0.35 \pm 0.17$ | $33.33 \pm 27.22$ | $\mathbf{100.00 \pm 0.00}$ |
| CIFAR-100 | $83.33 \pm 15.21$ | $82.29 \pm 15.05$ | $66.67 \pm 27.22$ | $\mathbf{100.00 \pm 0.00}$ |
| HAM10K | $0.52 \pm 0.43$ | $0.52 \pm 0.43$ | $1.56 \pm 0.74$ | $\mathbf{100.00 \pm 0.00}$ |

### E.2. Canaries at Different Metric Quantiles

In this section, we show a distribution of our metric scores and the canary samples and the attack results at 1% (our main result that produced the best canary examples), 5%, 10% and 25%.

### E.3. Effects of IF-Init

*Table 9.* Comparison of TPR@0.1%FPR on OptiFluence with IF-Init and with IF-Init replaced by ID initialization on MNIST and CIFAR-10.

| Dataset | Method | TPR@0.1%FPR |
|---|---|---|
| MNIST | OptiFluence | $100.00 \pm 0.00$ |
| MNIST | OptiFluence w/o IF-Init | $33.53 \pm 26.77$ |
| CIFAR-10 | OptiFluence | $99.60 \pm 0.19$ |
| CIFAR-10 | OptiFluence w/o IF-Init | $38.02 \pm 25.52$ |

We conducted ablation studies on CIFAR-10 in Section 6.3 (see Table 4), explaining our design choices at each stage. However, we had not directly compared the performance of OptiFluence with and without IF-Init. In Table 9, we report this comparison and observe a substantial gain in TPR@0.1%FPR when replacing random selection-based ID initialization with our IF-Init procedure. Specifically, TPR@0.1%FPR performance improves by 2.98x on MNIST and by 2.62x on

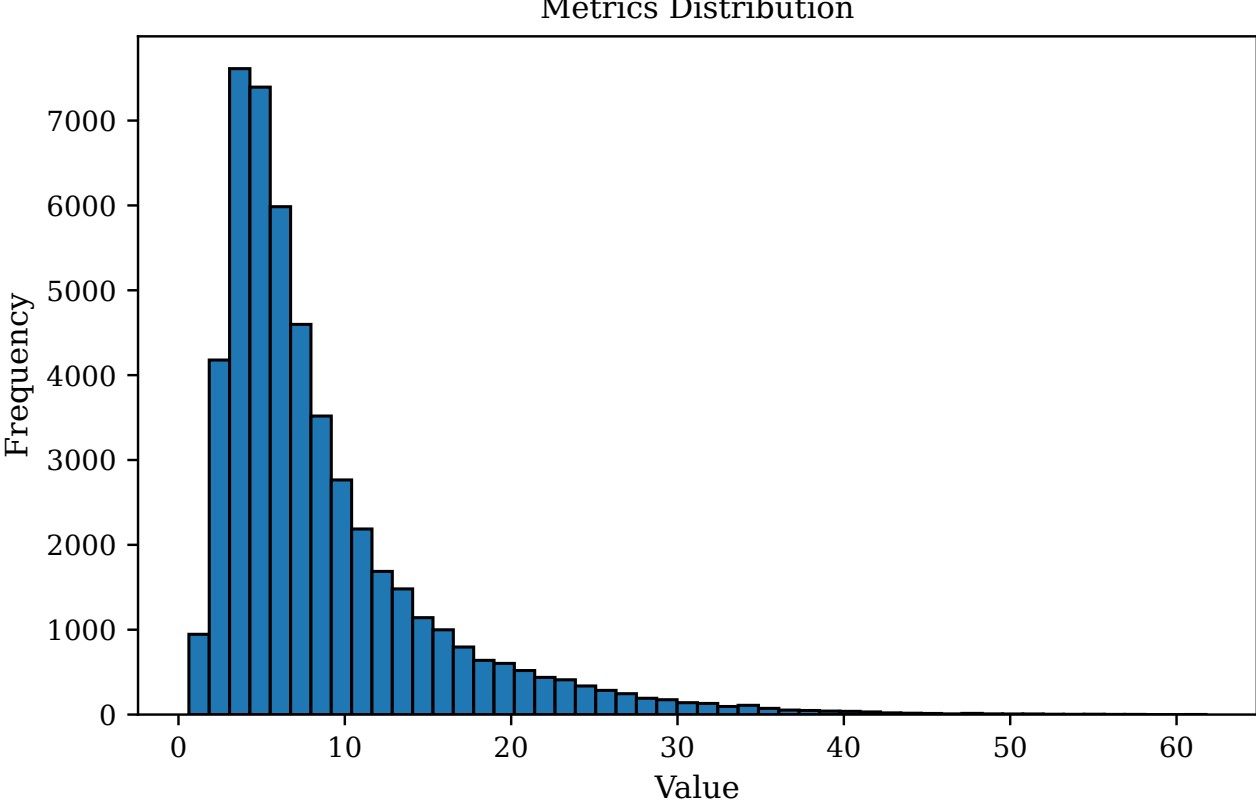

*Figure 4.* We show that the metrics of the training data follows a long-tail distribution on `CIFAR-10` dataset.

`CIFAR-10`, demonstrating the strong effectiveness of IF-Init.

### E.4. Qualitative Examples and ROC Curves

Figure 5 displays the optimized canaries from our main experiment (Table 1). On `MNIST`, the resulting canaries exhibit distinct patterns. For example, one unrolled optimized canary for `MNIST` (bottom left corner) resembles a negative of a two. In contrast, on `CIFAR-10`, the resulting canaries still largely resemble their initialization from real data. However, for one influence-optimized `MNIST` canary, the optimization procedure did not introduce any perceptible changes. The canary's ROC curve highlights that this is likely due to suboptimal optimization; the corresponding line in Figure 6 ("`MNIST` Influence-Optimized", green dotted) shows a consistently lower TPR.

### E.5. Canary Transferability

An important property of optimized canaries is transferability. Since generating canaries can be costly, one would hope that canaries optimized on a small model are still strong canaries for larger models. We hence optimize canaries for `MNIST` on the same MLPs as before, but then use those canaries to audit MLPs with larger widths. This corresponds to a threat model with a weaker adversary, where the adversary does not have perfect knowledge of the training procedure. Due to the large computational cost, we use the heuristic auditing scheme with 512 victim and 128 shadow models. For `CIFAR-10`, Figure 8(c) shows that canaries optimized on ResNet-9 transfer effectively to larger networks (ResNet-18, ResNet-50, WideResNet-16), with training configurations detailed in Table 7.

As seen in Figure 8 (a) and (b), canaries optimized on the smallest models are still strong canaries for larger models. In fact, we find that the TPR at 0.1% FPR even increases as model size increases. This hints that optimized canaries strongly exploit the memorization capabilities of a given model architecture, thereby generalizing beyond the specific model used for optimization. This property, in combination with recent work on efficient unrolling of SGD (Bae et al., 2024; Engstrom

et al., 2025), can enable more effective auditing of much larger models such as LLMs.

Figure 8 (c) highlights that influence-optimized canaries and mislabeled samples exhibit a similar scaling curve. However, Figure 8 (a) shows that unrolled canaries can achieve a very high TPR at low FPR even for very small widths, where the models' capacities are too small to memorize mislabeled samples.

### E.6. Distinguishability

**Optimization makes canaries more distinguishable.** We investigate how our optimization procedures make canaries more distinguishable. Concretely, we optimize a canary using both unrolled and influence-based optimization on an MLP for `MNIST`. We then take the canary after one step of optimization, after some intermediate steps, and at the end. For each, we fit 64 models with the canary in their training data and 64 models without. The results in Figure 7 show that our optimization procedure primarily affects models not trained on a canary; decreasing the LiRA hinge score for non-member models while retaining the score for member models.

### E.7. Impact of the Truncation Parameter $k$ in Truncated Backpropagation Through Time

In Figure 9, we show the canary loss during optimization for three different choices of k: 2, 4, and untruncated for canaries trained on our MLP model on `MNIST`. The latter is using our Unrolled baseline where no truncation occurs.

Despite these differences, we did not observe a meaningful impact of tuning $k$ in practice. Lower $k$ does not in fact impact the TPR@0.1%FPR scores of our attacks. As even canaries trained with $k = 2$ remain nearly perfectly detectable.

### E.8. Memorization in Mislabeled Canaries

The elevated in-distribution baseline visible in Table 1 (24.6% TPR on `CIFAR-10`) arises from strong memorization by ResNet-9 trained without regularization: standard SGD on this compact architecture memorizes a fraction of individual training samples, making some trivially detectable even without adversarial optimization. The subsections below characterize this effect and introduce a controlled configuration—augmentation and EMA—that reduces memorization pressure to recover discriminative power between canary strategies.

**Asymmetric vulnerability.** Evaluating LiRA on our canary variants revealed a striking asymmetry: on standard ResNet-9 (left panel of Figure 10), mislabeled canaries reach 72% TPR@0.1%FPR, adversarial examples reach 90%, and OptiFluence reaches 99.6%, while in-distribution canaries achieve only $\approx 25\%$. The vulnerability of mislabeled canaries is structural: a model trained with the mislabeled canary fits the incorrect label and develops high logit values ($\approx 8$ units) for the wrong class, while a model trained without it simply predicts the natural class. This gap—roughly 8 units for mislabeled canaries versus 0.48 units for correctly-labeled ones—is large enough that membership can be inferred trivially from logits alone, without the likelihood ratios that LiRA computes.

**Validity of the ResNet-9 baseline.** The ResNet-9 model trained without data augmentation or EMA is used throughout the main body as a deliberately minimal configuration. Its simplicity isolates the effect of canary optimization from confounds introduced by regularization: every algorithmic change to the optimizer is directly reflected in a measurable change in attack success. Moreover, the strong memorization signal—a consequence of standard SGD on a compact architecture with no generalization pressure beyond the task loss—maximizes the observable IN/OUT gap, making it straightforward to verify that a given canary is genuinely exploiting model vulnerabilities rather than exploiting noise.

**Augmentation and EMA for higher-resolution measurement.** To recover discriminative power and distinguish between canary strategies, we introduce per-epoch random crop (pad-4) and horizontal flip augmentation, together with an exponential moving average (EMA) of weights over the final epochs of training. Augmentation flattens the loss landscape around individual examples, reducing the model's ability to memorize specific pixel configurations. EMA, applied from a start fraction $f$ of training, produces an inference-time model that interpolates late-stage SGD iterates, smoothing away sharp minima associated with individual training points. Together, these techniques reduce the IN/OUT score gap to a range where differences between strategies manifest as measurable TPR variation rather than uniform saturation. As shown in the right panel of Figure 10, OptiFluence achieves 26.6% vs. 16.7% for IF-Init, 10.4% for mislabeled, and 3.6% for in-distribution canaries.

A key empirical finding concerns EMA hyperparameter sensitivity. The effective averaging window is $1/(1-d)$ epochs, where $d$ is the decay parameter. Setting $d = 0.99$ on a 20-epoch run produces a 100-epoch window: at epoch 5, the EMA is already 83% dominated by the near-random initialization, collapsing accuracy to around 10%. Using $d = 0.9$ (10-epoch window) with a start fraction of $f = 0.5$ achieves 55–70% accuracy while meaningfully reducing memorization. This sensitivity underscores that EMA hyperparameters are regime-specific and should be calibrated relative to total training duration, not transferred from longer training schedules.

The augmentation and EMA configuration is not a privacy defense; it is a controlled modification designed to bring our training regime into a range where attack success is neither trivially full nor trivially empty, providing a setting in which membership inference results carry informative signal about the vulnerability of specific canaries.

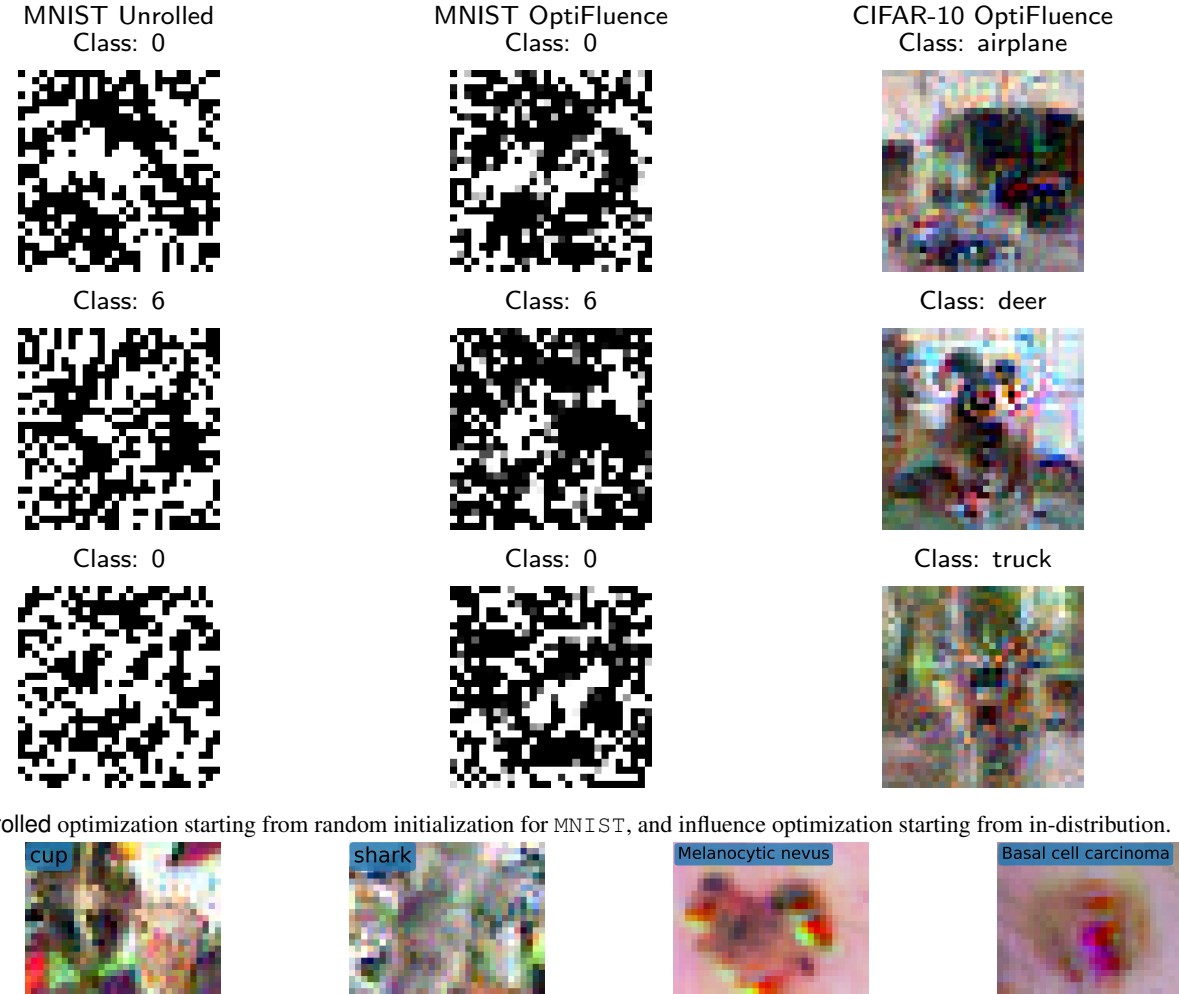

MNIST Unrolled
Class: 0

MNIST OptiFluence
Class: 0

CIFAR-10 OptiFluence
Class: airplane

Class: 6

Class: 6

Class: deer

Class: 0

Class: 0

Class: truck

*(a)* Unrolled optimization starting from random initialization for `MNIST`, and influence optimization starting from in-distribution.

cup

shark

Melanocytic nevus

Basal cell carcinoma

*(b)* OptiFluence-optimized canaries for two different seeds each on `CIFAR-100` (left) and `HAM10K` (right).

*Figure 5.* **Optimized canaries exhibit distinct patterns.** We show the canaries from Table 1. For Unrolled optimization on `MNIST`, starting from a random initialization, canaries often exhibit particular structure; for other methods and datasets, results are more subtle.

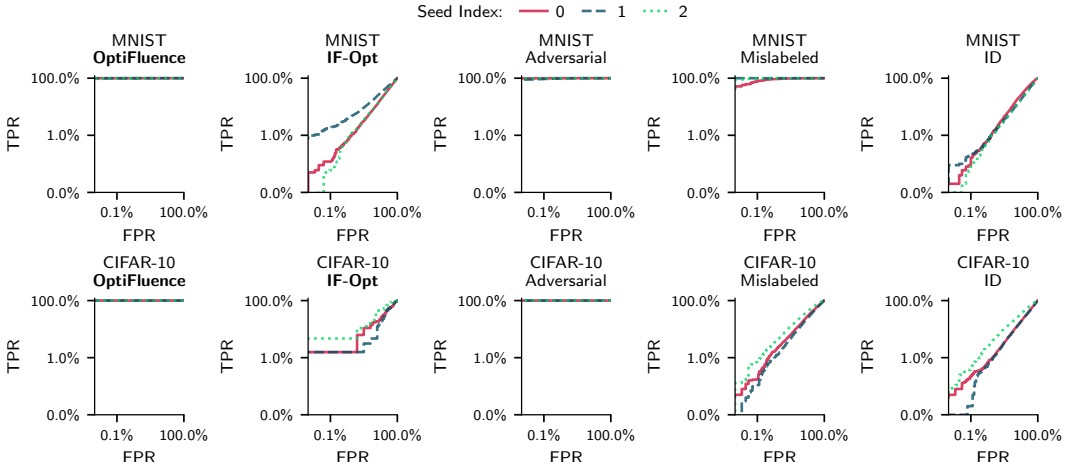

*Figure 6.* **ROC curves of optimized canaries.** We show all ROC curves of the canaries in Table 1. Different lines correspond to different seeds. Note that both the x-axis and y-axis are on a logarithmic scale.

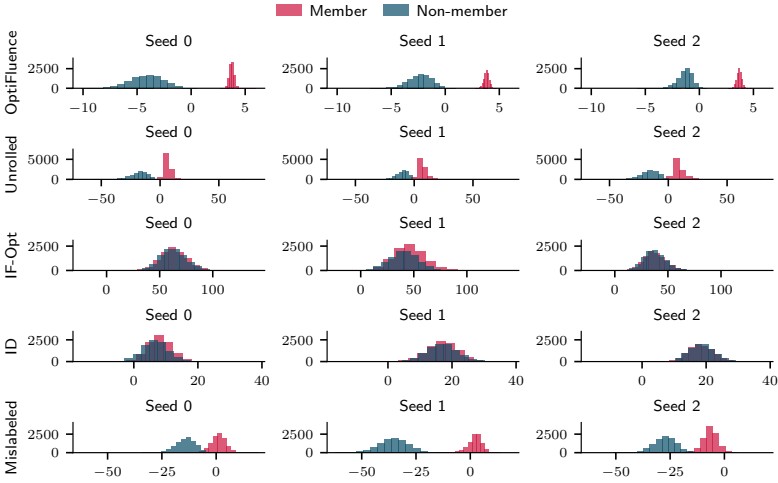

*Figure 7.* **Optimizing canaries reduces non-member scores.** We use intermediate canaries when using unrolled (top) and influence-based (bottom) optimization for a small `MNIST` MLP and train 64 models each with and without the canaries. Our optimization procedures primarily work by reducing the scores of non-member models.

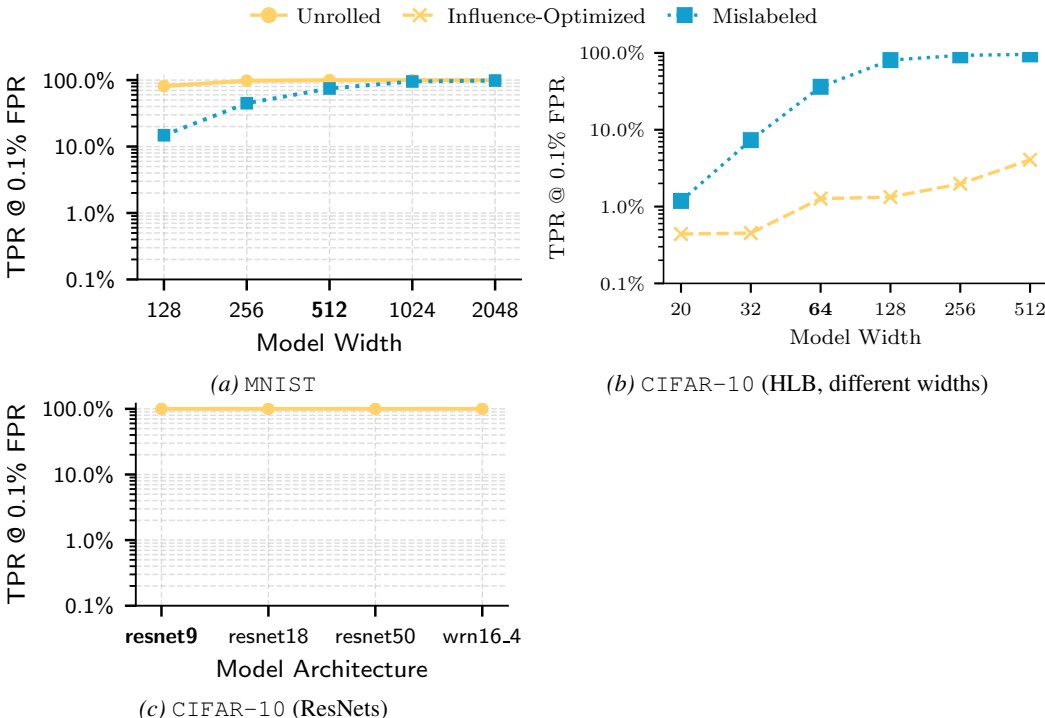

*(a)* `MNIST`

*(b)* `CIFAR-10` (HLB, different widths)

*(c)* `CIFAR-10` (ResNets)

*Figure 8.* **Transferability.** On `MNIST` (a), canaries from unrolled optimization achieve high TPR at low FPR even when the model width is too small to memorize mislabeled samples. For `CIFAR-10` (b), influence-optimized canaries also exhibit favorable transferability, but do not yet match the TPR of mislabeled samples at much larger models. For (c) we trained the models with a canary optimized on standard ResNet9 (no augmentation), and it consistently achieves perfect detectability.

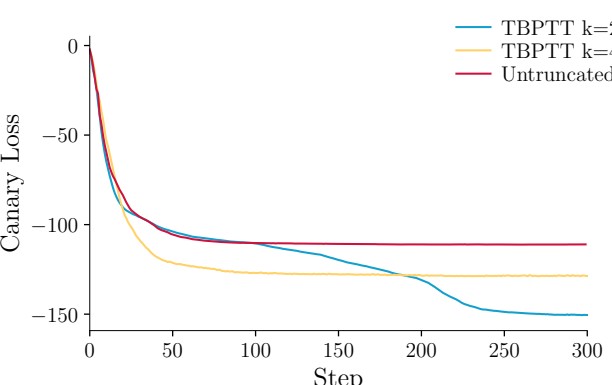

*Figure 9.* **Canary loss comparison between values truncation parameter $k$ on MLP models trained on `MNIST`.** In practice, all three canaries achieve nearly perfect detectability (99% TPR@0.1%FPR scores).

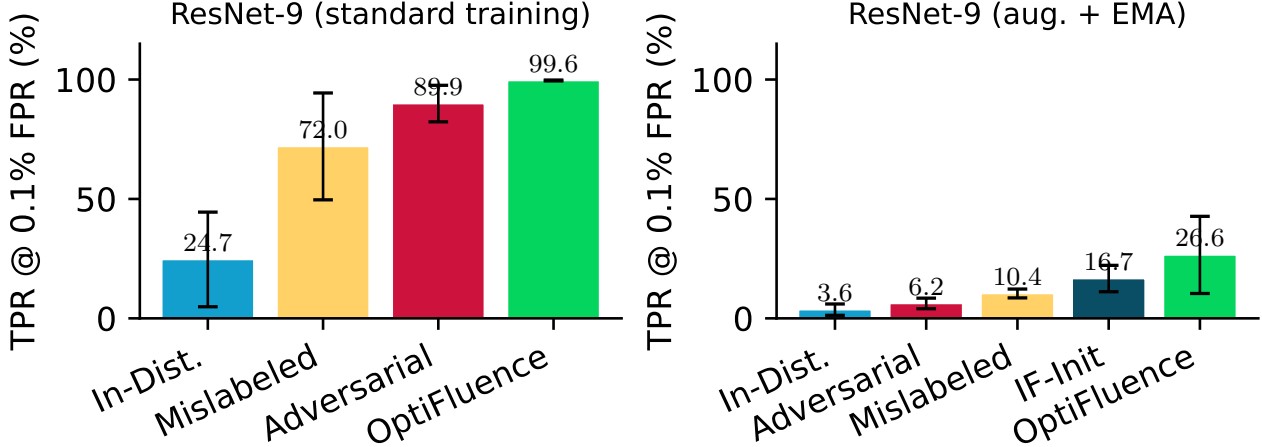

*Figure 10.* TPR@0.1%FPR for canary variants on ResNet-9 with standard training (left) and with augmentation and EMA (right). Standard training saturates at near-100% for most methods, preventing discrimination between strategies; augmentation and EMA reduce memorization pressure, making performance differences visible.

