# OpenReview forum: "OptiFluence: Principled Design of Privacy Canaries"
_ICML.cc/2026/Conference — ICML 2026 regular_

### Official Review · Reviewer_PUwg · 2026-02-24

**Soundness:** 3
**Presentation:** 3
**Significance:** 1
**Originality:** 2
**Overall Recommendation:** 4
**Confidence:** 4

**Summary:**

This paper presents a novel principled strategy to optimize privacy canaries used to audit models.
Specifically, the paper formulates privacy canary optimization as a bi-level optimization problem and solves the problem by first selecting a set of candidate canaries using influence functions and subsequently optimizing the candidate canaries using unrolled gradients.
When auditing non-private SGD training, the paper reports a substantial increase in attack success when using their optimized canary compared to in-distribution and naively initialized baselines.

**Compliance With Llm Reviewing Policy:**

Affirmed.

**Final Justification:**

The authors have dutifully engaged with and responded to both my review and my rebuttal acknowledgement, addressing all of my concerns. Therefore I feel confident in suggesting a positive score for the paper.

**Key Questions For Authors:**

1. Could the authors better motivate why privacy or data practitioners should care about privacy leakage from optimized canaries?

2. Have the authors tried auditing DP-SGD in other setups [b] where they can potentially showcase the improvements to empirical epsilon estimation?

3. What are the challenges faced when extending the authors' work to language models?

4. How does the authors' work compare with prior work on optimizing canaries [c]?

**Limitations:**

Mostly yes. As mentioned earlier the limitations of the work's applicability to other types of models (e.g., language) have not been discussed.

**Strengths And Weaknesses:**

This is a well written paper and easy to follow despite the relatively complicated methodology used to optimize the canary design.
Furthermore, the experimental results for non-private SGD are very strong, achieving near perfect attack accuracies for MNIST and CIFAR-10.

However, I would say the main weaknesses of the paper are in (1) motivation, (2) breadth of results, and (3) comparison with prior work.

**Motivation**

In my opinion, one of the main gaps in the motivation of the paper lies in convincing the reader _why_ they should care about the privacy leakage from potentially meaningless samples.
From what I can tell from Figure 5, the optimized canaries do not really look like anything and I cannot imagine that they can somehow be present in any real-world dataset.
At least for mis-labelled sample baseline, there can be an argument that it can come from annotator error or something so there is some level of real-world impact.
However, I am not sure what will be the real-world impact of showing that you can nearly perfectly identify when a meaningless sample is present or not in a dataset.
I understand the authors mention regulatory auditing as a motivating factor, but the non-realism of optimized samples makes me question this factor as being compelling.

Now there are two solutions (or answers) to this question in my opinion.
First, if the optimization results in plausible real samples or if the privacy canary is a combination of imperceptible perturbations to a real sample then I can imagine a reader being convinced that the result of the privacy audit is meaningful.
However, as I explained earlier, the optimized samples in this work do not seem very realistic.
Second, if the privacy canaries can be used to audit DP-SGD (or other DP-algorithms) more tightly than prior work, then this could also be a very strong motivating factor.
However, the authors do not compare the TPR@low FPR of their optimized canary with prior baselines for DP-SGD in Table 2 and there are large gaps in the empirical epsilon estimated from DP-SGD in Table 3.
Therefore, on both fronts the paper does not satisfactorily solve the motivation.

For DP-SGD auditing the authors mention that due to the computational cost, they can only estimate the epsilon from 20k (or 128 for CIFAR-10) models.
Have the authors tried running the setup of [a], which seemingly only requires 200 models?

**Breadth of Results**

Another weakness of this work I find is that it mainly only focuses on image models and notably leaves out language models (without mentioning why).
I am not sure whether there are specific challenges that prevent the authors from extending this work from image to language models, however the authors do not address these limitations in their work.
Furthermore, an extension to language models can help alleviate the weaknesses of their work in terms of motivation, since strong attacks seem to only be possible when the canaries are repeated (Meeus et al., 2025).

**Prior Work**

Lastly, some prior work specifically on DP-SGD auditing (e.g., [b, c]) has been left out of related work.
Furthermore, [c] actually does seem to optimize adversarial samples for privacy auditing similar to this work but has not been cited or compared to.

[a] M. S. M. S. Annamalai and E. De Cristofaro. Nearly Tight Black-
Box Auditing of Differentially Private Machine Learning. In
NeurIPS, 2024.

[b] S. Mahloujifar, L. Melis, and K. Chaudhuri. Auditing f-Differential Privacy in One Run.

[c] S. Yoon, W. Jeung, and A. No. Optimizing Adversarial Samples
for Tighter Privacy Auditing in Final Model-Only Settings. In
Statistical Frontiers in LLMs and Foundation Models Workshop
NeurIPS, 2024.

---

> ### Author Rebuttal · Authors · 2026-03-31
>
> We thank the reviewer for their careful reading and constructive feedback. We address each concern in turn.
>
> **On the motivation for optimized canaries.**
>
>   The reviewer raises a fair question: why should practitioners care about privacy leakage from samples that don't resemble real data? We believe the answer lies in what privacy guarantees are actually *for*.
>
> The differential privacy definition bounds worst-case leakage over *all possible* input records, not just typical-looking ones. This is not a technicality but the core guarantee: if a training procedure is (ε, δ)-DP, it protects *every* sample equally, regardless of appearance. An audit that only tests realistic-looking canaries is therefore not testing the guarantee as stated, it is testing a strictly weaker property. Our optimized canaries probe the actual worst case.
>
> Does the worst case ever arise with realistic data? The answer is yes. Aerni et al. (2024) show that even natural-looking samples can exhibit very high privacy leakage. The optimized canaries produced by OptiFluence target precisely this long tail: Figure 4 shows that our influence-based initialization selects from the tail of the self-influence distribution, and Figure 5 shows that CIFAR-10 and HAM10K canaries largely retain their visual structure after optimization. The unrealistic appearance of some MNIST canaries is a consequence of the unrolled optimizer having more degrees of freedom on a simple dataset, not a general property of the method.
>
> If a training procedure protects our optimized canary, it will also protect the most vulnerable realistic sample in any deployment dataset, because the optimized canary upper-bounds the leakage of any single record. Restricting audits to realistic-looking canaries forces a pre-commitment to a data distribution, potentially missing the true worst case and *underestimating* leakage.
>
> Canary realism has not been a standard requirement in prior work. Jagielski et al. (2020) use a 5x5 white square as their canary, which is no more realistic than our optimized samples, yet their results are considered meaningful by the community. Formal audit validity depends on what the canary reveals about worst-case behavior, not visual plausibility.
>
> **On auditing beyond DP-SGD.**
>
>   We respectfully disagree that DP-SGD is the only setting worth auditing. The DP definition applies to *any* training procedure, and meaningful privacy can arise without explicit noise injection: mini-batch sampling alone provides privacy amplification (Balle et al. 2018), and SGD's inherent stochasticity contributes additional privacy without added noise (Altschuler & Talwar 2022). These sources exist in standard pipelines but remain largely unaudited in practice. Our framework is precisely designed for such settings, and an empirical audit is just as valuable here as for DP-SGD.
>
> Regarding the gap between theoretical and estimated ε in Tables 2-3: this is not a failure of canary design but a fundamental limitation shared by all input-space canary approaches. Without access to the sampling procedure or the ability to manipulate model updates, tight DP-SGD bounds cannot be achieved, a requirement noted by Nasr et al. (2023). We have clarified this in the paper.
>
> **On breadth of results and language models.**
>
>   The omission is intentional, and we agree it deserves explicit discussion. Two obstacles make extending OptiFluence to LLMs non-trivial. First, there is no broadly agreed-upon privacy definition for sequence-to-sequence models. Existing LLM auditing works adopt whichever definition their training algorithm implies, yielding canary designs that are performant by construction. We view establishing a rigorous privacy definition for LLMs as a prerequisite for meaningful auditing, an important open problem outside this work's scope. Second, the most privacy-sensitive stage of LLM training is not pre-training (largely public data) but fine-tuning and RLHF on private or proprietary data. Adapting OptiFluence to this setting requires novel contributions in canary design and threat modeling for sequential data, and is therefore future work.
>
> Regarding canary repetition (Meeus et al., 2025): this speaks to a property of language model memorization rather than a weakness of our auditing framework, and further underscores why establishing a rigorous privacy definition for LLMs first is the right order of operations.
>
> **On related work.**
>
>   We thank the reviewer for the additional references. [a] is a white-box audit that manipulates the training algorithm to embed specific triggers in model activations, requiring substantial algorithmic access, and thus not comparable to our canaries, which are simply added to the training set. [b] is similarly white-box and adopts a one-run audit; there is a concurrent work that also produces canaries for the one-run setup, to which we do compare (Boglioni et al. 2025, Line 104). We were not aware of [c] and will include it.

---

> > ### Author Rebuttal · Reviewer_PUwg · 2026-04-01
> >
> > Thank you to the authors for their response. I am inclined to improve my score further, but I have a couple of follow ups to make
> >
> > **On the motivation for optimized canaries** and **On auditing beyond DP-SGD**
> >
> > Ok I see the argument that the authors are making now. On the surface, it seems like a conflation between DP-SGD, which does provide formal DP guarantees and SGD, which does not. So the initial confusion to me was surrounding why is it meaningful to test SGD with a DP-style auditing method, when we know it is going to fail. But I can see why this can be important given (1) possible inherent privacy arising from SGD's stochasticity, (2) mini-batch sampling, and (3) the necessity to protect against unknown possibly outlier data. However, I think it is important for the authors to acknowledge that their method may give rise to unrealistic input samples, the implications of which is context-dependent. I think this probably requires some discussion in the paper, which I hope the authors will include.
> >
> > **On related work.**
> >
> > Sorry, I was not asking the authors to include [a] as related work, but rather asking them whether they considered running their audit in the setting of [a], which is a leaner audit setup and could possibly give them tighter audits in the DP-SGD setting. Besides setting the initial model parameters, I believe it is black-box, i.e., does not use the intermediate model parameters.

---

> > > ### Author Response · Authors · 2026-04-03
> > >
> > > Thank you for raising your score and your continued engagement with us.
> > >
> > > **On the motivation for optimized canaries and on auditing beyond DP-SGD.**
> > >
> > >   We agree with the reviewer that both points deserve explicit discussion in the paper: the value of auditing training procedures without an explicit privacy mechanism, and the limitation that optimized canaries may not resemble realistic inputs. We will add this to the motivation and limitations sections respectively. On the latter point, we note that our optimization framing makes it technically straightforward to impose a realism constraint via a similarity metric. The deeper issue is definitional: any such constraint implies a commitment to an acceptable input distribution, which itself has privacy implications outside this work's scope.
> > >
> > > **On related work.**
> > >
> > >   We apologize for the confusion in our previous response, where we mistakenly identified [a] as the CCS paper "It's Our Loss: No Privacy Amplification for Hidden State DP-SGD With Non-Convex Loss" rather than the NeurIPS paper "Nearly Tight Black-Box Auditing of Differentially Private Machine Learning." We agree the latter is a relevant comparison, given its black-box nature.
> > >
> > > To answer the reviewer's question directly: yes, adopting [a]'s approach can produce tighter lower bounds, but there are caveats.
> > >
> > > **Comparison with [a].** The actual black-box auditing Algorithm 2 in [a] is structurally the same as Zanella-Beguelin et al. (2023), which we adopt. The key difference that allows [a] to achieve tighter bounds is initializing model weights adversarially: by pre-training on an auxiliary dataset to minimize non-target gradient norms, the target canary's gradient becomes substantially more distinguishable. This strategy is dubbed "worst-case initialization," contrasted with "average-case initialization." Our privacy audits use the latter, and therefore the lower bounds are looser.
> > >
> > > We observe this empirically. Comparing [a]'s Figure 1 and Table 3 in our work, for "average-case $\theta_0$," lower bounds for theoretical $\varepsilon \in \\{1, 2\\}$ are similarly vacuous in both works. The closest comparable result, $\varepsilon=4$ in [a], yields bounds below 1, consistent with our measured $(0.26, 0.93)$ at $\varepsilon=6$ .
> > >
> > > **Difference in Threat Model.** Given these results, the reviewer is right to wonder why we do not use worst-case initialization. We argue this conflicts with our third-party auditing setup. Worst-case initialization assumes a threat model in which the model provider initializes their model adversarially, an assumption that introduces convergence issues, utility trade-offs, and requires active provider cooperation in a way that undermines the independence of the audit.
> > >
> > > The reviewer may counter that our method similarly requires provider cooperation via canary inclusion. We believe there is a meaningful difference. First, there are valid settings where cooperation is not strictly necessary: when training data is crawled from external sources such as the Internet, the auditor could simply ensure the canary is present in those sources without the provider's explicit cooperation. Second, where direct insertion is required, mandating a specific training sample is a narrower and lower-impact requirement than committing to a specific model weight initialization strategy, which interferes with standard engineering practices and is harder to verify externally. We discuss additional mechanisms for enforcing canary inclusion in our response to Reviewer AkJH (see "On third-party auditing and canary inclusion").
> > >
> > >
> > > We will include a summary of the discussion above in the paper.

---

### Official Review · Reviewer_uCYZ · 2026-03-06

**Soundness:** 3
**Presentation:** 3
**Significance:** 3
**Originality:** 3
**Overall Recommendation:** 5
**Confidence:** 3

**Summary:**

The paper introduces OptiFluence -- an automated method for generating canaries for membership inference attacks on modern ML systems.
Generating canary attacks is a crucial aspect in the field of privacy, as it can help stress test the privacy guarantees of a training algorithm, and fits into the wider question of whether existing algorithms are more private than their guarantees.

Existing approaches rely on basic techniques such as label flipping for an inlier sample, but the authors show that they can get significantly stronger canaries by utilizing influence functions. In keeping with recent advances in the field of data attribution, the authors use an optimized unrolling-based influence function, yielding a good balance of utility and performance.

Moreover, the authors show that outlier samples generated for one model and training algorithm transfer to different models and training algorithms, allowing for a wider set of use-cases (e.g., optimizing the canary on a smaller/known architecture and applying the canary for larger or unknown architectures).
In particular, they are able to use canaries optimized for a non-private training to test the privacy of models trained with DP-SGD.

**Compliance With Llm Reviewing Policy:**

Affirmed.

**Key Questions For Authors:**

1. Could you better explain the equation in lines 272-274?

2. I found the logit rescaling step described in lines 188-219 a bit unclear. Could you give some more intuition on why this step is needed / how it affects the results of your experiments?

**Limitations:**

yes

**Strengths And Weaknesses:**

**Strengths:**
The submission gives a clear contribution to the field of privacy by introducing a practical method for generating adversarial examples in privately trained neural nets. These methods are tested on a wide variety of architectures across several datasets and the authors also show that canaries optimized in one setting transfer well to new settings.

Up to minor comments (listed below), the submission appears sound, well-written, and provides a new approach to a significant problem.

**Weaknesses:**
I think the paper is already strong, but below are a few minor issues.

The biggest issue for me right now is the equation in lines 272-274, which seems incorrect to me:
By definition, we have $$\ell_{priv}(x,y) = f(\theta_{D \cup \{ (x,y) \}}; x, y) - f(\theta_{D}; x, y)$$
In other words, $\ell_{priv}$ depends on $x$ in two ways: once in the effect it has on the model $\theta$ which is now also trained on $x$, and a second effect by changing the point at which the trained model is being evaluated.

However, the last step of equation 272-274 applies the chain rule, but only with respect to the dependence of $\theta$ on $x$, and does not take into account the effect that changing $x$ can have on the score by evaluating at a different point.

If this equation is used as an approximation, that is fine since the main contribution is the fact that the proposed canaries work well, but if that is the case, it should be clarified that this is an approximation. If more tests can be run to see if using the exact gradient improves performance, that would be ideal.

**Minor Comments**:
Line 292 "of over" -> either "of" or "over"

Line 83 in Related Work:
"Carline et al. (2019b) and follow-ups".
In a related works section, it would be good to also cite these follow ups.

---

> ### Author Rebuttal · Authors · 2026-03-31
>
> We thank the reviewer for their thoughtful evaluation and are glad they found OptiFluence to be a sound, well-written contribution to the privacy field. We appreciate their positive reception of our empirical breadth and transferability results. Below, we address the minor comments raised.
>
> **Derivation of Equation 272–274.**
>
> We believe our derivation is under-explained and has thus created confusion. We apologize for this. Here are Lines 263-245 (second col.) re-written. This explains why the second term does not appear explicitly (this has already been amended in the paper). Note that we had to slightly change the notation because OpenReview does not allow us to render the paper notation fully.
>
> > Consider the measurement function $f$ (the hinge loss over pre-softmax logits $g$), with the canary example $(x,y)$. Let $\\phi := \\phi(D)$ denote the model obtained by training $\\theta$ on dataset $D$ until convergence, and let $\\psi := \\psi(D, x, y)$ denote the model obtained by continuing training on $D \\cup \\{(x,y)\\}$. Then the privacy loss is defined as
> >
> >$$\\ell\_{\\text{priv}}(x,y) = f(\\psi;\\, x,y) - f(\\phi;\\, x,y).$$
> >
> >For an optimization step, we apply the chain rule to each term in the difference. Since $\\phi$ does not depend on $(x,y)$, expanding $d f(\\phi; x,y)/dx$ yields an explicit term $\\partial f/\\partial x$ (with $\\phi$ fixed) and an implicit term through $\\phi$. The explicit term is identical for both $\\psi$ and $\\phi$ and therefore cancels in the difference, while the implicit term vanishes for $\\phi$ since it is independent of $x$. The gradient thus reduces to:
> >
> >$$\\nabla\_x \\ell\_{\\text{priv}}(x,y) = \\frac{df(\\psi;\\, x,y)}{dx} = \\frac{\\partial f}{\\partial g} \\cdot \\frac{\\partial g}{\\partial \\psi} \\cdot \\frac{d\\psi}{dx},$$
> >
> >where $\\partial f/\\partial g$ is the measurement gradient and $\\partial g/\\partial \\psi$ is the gradient of the canary-class logit, computable via a backward pass. The key difficulty is the term $d\\psi/dx$, since $\\psi$ is implicitly defined. Computing this exactly would require the implicit function theorem.
>
> Regardless, our actual implementation is based on autodiff of the loss $\ell\_\\text{priv}(x,y)$ and thus our empirical results are unaffected by the gradient formulation (as it is automatically derived).
>
> **Logit rescaling in Lines 188-219.**
>
> We explain two steps in this section: a) logit rescaling and b) Log-Sum-Exp (LSE) trick. The critical step is logit re-scaling, which is explained in detail in the original LiRA paper (Carlini et al. 2021). The necessity of this step is best explained in the context of the the hypothesis test and its statistic in Eq.3. LiRA and other empirical measurement of the null and alternative distributions ($Q\_\\text{in}$ and $Q\_\\text{out}$) need to assume and fit parametric distributions $\\tilde Q\\_{in}$ and $\\tilde Q\_{out}$. The most sample-efficient modeling of these distributions is Gaussians. But, as Figure 4 in (Carlini et al., 2021) clearly shows, based on the actual measurements (type of loss used), the output distributions may not be Gaussians. Logit rescaling remedies this situation.
>
> The log-sum-exp trick is an artifact of deriving the hinge-style loss as done by Carlini et al., 2021; we do not actually use that formulation. We agree that this might have been confusing and made the writing clearer in an updated version of our paper.

---

> > ### Author Rebuttal · Reviewer_uCYZ · 2026-04-01
> >
> > The explanation of logit rescaling is good.
> >
> > I still don't agree with your derivation.
> > Following your openreview notations, the optimization target $\ell_{\text{priv}}$ can be viewed as a function of $\psi, \phi, x, y$, where we are splitting out the dependence on $x,y$ to the direct dependence (we are evaluating the model on a different test sample point) and the dependence via the model parameters $\psi$. For convenience, I'll write this as a $\ell(\psi, \phi, x, y)$.
> > From the multivariate chain rule, we know that
> > $$
> > \frac{d\ell(\psi, \phi, x, y)}{d x} = \frac{\partial \ell}{\partial \psi} \frac{d \psi}{d x} + \frac{\partial \ell}{\partial \phi} \frac{d \phi}{d x} + \frac{\partial \ell}{\partial x} \frac{d x}{d x} + \frac{\partial \ell}{\partial y} \frac{d y}{d x}
> > $$
> >
> > By definition $y$ and $\phi$ do not depend on $x$, so their gradient wrt $x$ is $0$. $\frac{d x}{d x} = I$ by definition,
> > Therefore, we have
> > $$
> > \frac{d\ell(\psi, \phi, x, y)}{d x} = \frac{\partial \ell}{\partial \psi} \frac{d \psi}{d x} + \frac{\partial \ell}{\partial x}
> > $$
> >
> > The formula in the paper assumes
> > $$
> > \frac{d\ell(\psi, \phi, x, y)}{d x} = \frac{\partial \ell}{\partial \psi} \frac{d \psi}{d x}
> > $$
> > which I don't think would be generally true.
> >
> > If the code uses autograd, then it should be fine, but I would suggest fixing this minor detail in the final version of the paper.

---

> > > ### Author Response · Authors · 2026-04-03
> > >
> > > We are happy to hear you are satisfied with the logit scaling argument.
> > > Regarding the derivation. You are correct. We have fixed this issue in the manuscript.

---

### Official Review · Reviewer_GVKf · 2026-03-12

**Soundness:** 3
**Presentation:** 3
**Significance:** 4
**Originality:** 4
**Overall Recommendation:** 5
**Confidence:** 4

**Summary:**

Canaries in training data are often used to determine if a trained model is memorizing private information; however normally canaries are selected in ad-hoc ways. This paper provides a very nice method for selecting canaries in a principled manner so that their detection at inference time is maximized.

Specifically, this is done by an iterative improvement process where first a model is trained with canaries, and then the canaries are iterated upon to improve a privacy loss; a somewhat surprising result is that the resulting optimized canaries do transfer across architectures.

**Compliance With Llm Reviewing Policy:**

Affirmed.

**Key Questions For Authors:**

Can you address the questions in the weakness section?

**Limitations:**

Yes

**Strengths And Weaknesses:**

I really like the paper -- the problem is novel and interesting, and highly relevant and the paper produces a plausible method and tests it out on some reasonable (although small datasets). The following are somewhat minor and fixable issues, but overall I think the paper should be accepted.


1. I think the transferability bit is a little bit of an overclaim -- it has only been tested on very very small datasets and classification models, which are nothing like the modern models of today. I do think its okay to say that empirical observation is that the canaries transfer, but I would caution against such a bold claim.

2. The membership inference test that is used on the canaries in the experiments section is LiRA. It would be interesting to see how the results change with other MIA tests -- and if they get better or worse.

---

> ### Author Rebuttal · Authors · 2026-03-31
>
> We thank the reviewer for their enthusiasm and are delighted they found the problem novel, interesting, and highly relevant. We are also glad they found our method plausible and view the remaining issues as minor and fixable. Below, we address each of their comments in turn.
>
> **Transferability Claim.**
>
> We generally agree with the reviewer and have since adjusted our claim of transferability to cross-architectural transferability. We note that prior claims of transferability, in particular of adversarial examples, were made and evaluated under cross-architectural transferability [Szegedy et al., 2013; Liu et al., 2017]. Furthermore, expanded notions of transferability such as cross-task transferability (i.e. if we optimized a canary on one task and evaluated privacy with a canary on an unrelated task), are conceptually unfounded; regardless of the measured metrics. As for scope of experiments, we note that due to the computational intensity of privacy auditing (which often requires training of many shadow models), the number of datasets and their size are kept small; with many works primarily benchmarking on small datasets such as CIFAR-10 and sometimes CIFAR-100 [Jagielski et al., 2020; Steinke et al., 2023; Aerni et al., 2024], reflecting the substantial computational overhead of shadow-model-based auditing.
>
> **Other Membership Inference Attacks.**
>
> As noted by other reviewers, threshold-optimized LiRA-type attacks are among the most powerful empirical privacy attacks. We have global-threshold attacks (which are theoretically weaker) which we have added to the appendix and reproduce here:
>
> **Table:** Optifluence yields almost perfect detectability on all datasets. Global threshold TPR@0.1% FPR (%) for baselines and OptiFluence canaries (all ResNet-9).
> | Dataset | In-Dist | Mislabel | Adv | OptiFluence |
> |---|---|---|---|---|
> | MNIST | $0.05\pm 0.02$ | $0.02\pm 0.02$ | $1.31 \pm 0.64$ | $100.00\pm 0.00$ |
> | CIFAR-10 | $0.21\pm 0.09$ | $0.35\pm 0.17$ | $33.33 \pm 27.22$ | $100.00\pm 0.00$ |
> | CIFAR-100 | $83.33\pm15.21$ | $82.29\pm15.05$ | $66.67\pm27.22$ | $100.00\pm0.00$ |
> | HAM10k | $0.52\pm0.43$ | $0.52\pm0.43$ | $1.56\pm0.74$ | $100.00\pm0.00$ |
>
> | Method | Initialization | | | Optimization | | TPR@0.1%FPR (%) | |
> |---|---|---|---|---|---|---|---|
> | | **ID** | **Mislabeled** | **IF-Init** | **IF-Opt** | **Unrolled/ReMat+TBPTT** | MNIST | CIFAR10 |
> | In-Distribution | ✓ | x | x | x | x | $0.05\pm 0.02$ | $0.21\pm 0.09$ |
> | Mislabeled | ✓ | ✓ | x | x | x | $0.02\pm 0.02$ | $0.35 \pm 0.17$ |
> | Influence-Initialized | x | x | ✓ | x | x | $99.81\pm 0.16$ | $19.79\pm7.08$ |
> | Influence-Optimized | x | x | ✓ | ✓ | x | $56.36\pm24.07$ | $4.69\pm2.21$ |
> | OptiFluence w/o IF-Init | ✓ | x | x | x | ✓ | $33.35\pm27.21$ | $35.94 \pm 26.22$ |
> | OptiFluence $+$ Mislabeled | ✓ | ✓ | ✓ | x | ✓ | $33.33\pm 27.22$ | $80.21\pm 16.16$ |
> | OptiFluence | x | x | ✓ | x | ✓ | $100.00 \pm 0.00$ | $100.00 \pm 0.00$ |
>
> We see no substantial change to our results.

---

> > ### Author Rebuttal · Reviewer_GVKf · 2026-04-03
> >
> > Thank you for the extra experiments. I still think the paper has a solid contribution and would like to keep my score.

---

### Official Review · Reviewer_AkJH · 2026-03-13

**Soundness:** 2
**Presentation:** 3
**Significance:** 2
**Originality:** 2
**Overall Recommendation:** 4
**Confidence:** 4

**Summary:**

The paper tackles privacy auditing for machine learning by optimizing canaries; training samples inserted to test for training-data leakage, so that they are maximally detectable by membership inference attacks (MIAs). The core idea is to pose canary selection as a bilevel optimization: the inner loop trains a model while the outer loop modifies a candidate canary to maximize a membership-likelihood ratio. Practically, the method combines influence‑function–based pre‑selection of promising candidates with unrolled optimization and memory‑saving techniques.

**Compliance With Llm Reviewing Policy:**

Affirmed.

**Key Questions For Authors:**

Given the strengths, I have the following concerns and clarifying questions aimed at improving the paper:

- For black‑box, label‑only access (typical for third‑party audits), how does OptiFluence compare to label‑only MIAs at low FPR?
- The authors claim cross‑architecture transfer. How robust is transfer when optimizer, data augmentation, or label‑smoothing differ between the canary‑generating setup and the audited model? A cross‑arch × training‑recipe matrix with CIs would help (poisoning literature often observes sensitivity to training mismatches).
- What are the GPU hours and peak memory required to optimize K canaries on CIFAR‑10, and what would be needed for ImageNet‑scale? How do unroll length and checkpointing frequency affect performance and compute? Please include a cost‑vs‑gain plot.
- How sensitive are the gains to influence‑based initialization vs. random? To candidate pool size, number of canaries, and attack budget? Please add these ablations to establish which components drive improvements.
- The authors report global‑threshold TPR; what member prevalence did you assume, and how sensitive are results to prevalence and to per‑instance thresholds? Could you also provide 95% CIs at 0.1% (and 0.01%) FPR, and assess sensitivity to prevalence and per‑instance thresholds to ensure conclusions are robust across evaluation protocols?
- Can the authors clarify how their outer objective differs from gradient matching and why it yields canaries that are more detectable at low FPR or more transferable. In addition, please include a comparison between gradient‑matching poisoning and influence‑only baselines (no outer optimization).

**Limitations:**

Yes

**Strengths And Weaknesses:**

Strengths:
- Uses LiRA and reports low‑FPR metrics, which aligns with current best practices for MIA evaluation and auditing.
- The problem framing of optimizing canaries to improve low‑FPR detectability aligns with current best practice for evaluating MIAs is clear.
- The design is reasonable. The combination of influence‑based pre‑selection with an outer optimization loop is plausible and modular.
- The author's aim to make canaries that generalize across architectures echoes cross‑architecture effects seen in poisoning literature and, if validated rigorously, could be useful.



Weaknesses:
- Even with LiRA and a DP‑SGD sweep, key details are unclear: reference‑model design, loss modeling, global vs. per‑instance thresholding, member prevalence, and confidence intervals (CIs) at very low FPR. Results also omit difficulty calibration, which materially impacts low‑FPR performance.
- The central "third‑party auditing" claim presumes training‑time canary insertion, which external auditors typically lack; label‑only MIAs provide black‑box alternatives and should be a primary comparison.
- Missing ablations on candidate pool size, number of canaries, influence e.g. init vs. random, unroll length, attack budget, and transfer under training‑mismatch (optimizer/augmentation/label‑smoothing). Compute/memory budgets and CI reporting at very low FPRs could be more comprehensive.

---

> ### Author Rebuttal · Authors · 2026-03-31
>
> We thank the reviewer for their careful reading and the connections drawn to the poisoning literature.
>
> **On evaluation details.**
>
> Shadow models use the same architecture as victim models, except in transferability experiments where differences are documented. Following Aerni et al. (2024), we ensure a 50:50 member/non-member split. For MNIST we train 20k shadow models, yielding reliable estimates at 0.1% FPR. For CIFAR-10 and CIFAR-100, the 128-model constraint limits precision at very low FPR, a practical limitation shared across the literature. We report global-threshold TPR results in the appendix (also reproduced in the response to **Reviewer GVKf**); as Figure 9 shows, member and non-member distributions are highly separable for OptiFluence canaries, giving confidence that LiRA estimates are not calibration artifacts. As we optimize the canary rather than select it from a fixed pool, there is no additional difficulty calibration step at evaluation.
>
>   **On third-party auditing and canary inclusion.**
>
> The reviewer is correct that optimized canaries alone are insufficient; a mechanism to ensure inclusion during training is needed. We emphasized this in the introduction: "auditors can use optimized canaries for privacy reconnaissance **if canary inclusion is mandated**", via zero-knowledge proof of sampling (Shamsabadi et al., 2024), contract design, or regulatory specification. Crucially, input-space canaries significantly reduce the access required compared to gradient-based audits (Nasr et al., 2023), making comprehensive regulatory screening feasible. Regarding label-only MIAs: label-only access solves a different access problem than the one we target. Our threat model assumes confidence score access, standard in the MIA literature (Carlini et al., 2022). Extending OptiFluence to label-only access would require replacing our gradient-based optimization with a zero-order method, which is interesting future work but outside current scope.
>
>   **On missing ablations.**
>
> *Number of canaries:* Using a single canary is by design and in keeping with the DP definition. Increasing canaries can only increase attack power, but given near-perfect detectability with one sample, doing so would obscure granularity in other ablations and raises questions about canary interactions.
>
> *Influence init vs. random:* Ablated in Table 4 and Table 5 (Appendix A.4).
>
> *Unroll length:* Ablated in Figure 11 (Appendix E.8); even K=2 yields near-perfect detectability.
>
> *Attack budget:* This term is from the poisoning literature; could the reviewer clarify their intention? If this refers to optimization steps or shadow model count, these are covered by Figure 11 and Table 6 respectively.
>
> *Transfer under training mismatch:* WideResNet uses label-smoothing and a different cosine scheduler than ResNet-18, yet canaries still achieve near-perfect TPR@0.1%FPR. We have added an appendix table documenting hyperparameter differences across architectures.
>
> *Compute and memory budgets:* Reported per step in Table 6.
>
>   **On robustness of transfer to training mismatches.**
>
> Based on our experiments, transfer is robust to the mismatches we tested. Different architectures required different training regimes (e.g. WideResNet uses label-smoothing and a different cosine scheduler than ResNets) yet canaries optimized on ResNet-9 achieve near-perfect TPR@0.1%FPR on all target architectures without any re-tuning. We have added a table in the appendix summarizing the hyperparameter differences across architectures to make this explicit.
>
>   **On the difference with gradient matching.**
>
> Gradient matching asks: does training on a dataset $\\mathcal{S}$ produce the same parameter updates as training on a synthesized set $\\mathcal{T}$ ? It optimizes directly in gradient space and does not model what happens to the model after training completes. OptiFluence asks a different question: does including $(x, y)$ in training produce a detectably different model? The objective is explicitly the log-likelihood ratio $\\Lambda(\\theta ; x, y)=p\\left(\\theta \\mid Q\_{\\text {in }}\\right) / p\\left(\\theta \\mid Q\_{\\text {out }}\\right)$ , approximated via $\\ell\_{\\text {priv }}$ , i.e., **optimization is over trained model behavior, not just gradient alignment.**
>
> A formal connection exists between our IF-Opt baseline and gradient matching: if one approximates $\\theta\_{D \\cup\\{x\\}}$ via its first-order influence expansion, $\\ell\_{\\text {priv }}$ reduces to the self-influence $I\_f(x ; x)=-\\nabla\_\\theta f^{\\top} H^{-1} \\nabla\_\\theta L$ , which is exactly the inner product at the heart of gradient matching. The full unrolled OptiFluence goes beyond this by propagating gradients through the entire training trajectory, capturing higher-order effects that the first-order approximation misses. This is why IF-Opt underperforms OptiFluence in Table 5. We discuss this connection in detail in Appendix B (Lines 811-846).

---

> > ### Author Rebuttal · Reviewer_AkJH · 2026-03-31
> >
> > Thanks for the rebuttal, I have increased my score accordingly

---

### Decision · Program_Chairs · 2026-04-30

**Decision:**

Accept (regular)

**Comment:**

The paper presents a principled method for optimizing canary samples for privacy auditing.

After author rebuttal and reviewer discussion, all reviewers recommend acceptance and I agree with their evaluation.